# Circulation of *Salmonella* spp. between humans, animals and the environment in animal-owning households in Malawi

Catherine N. Wilson [1,2,3,4] ✉, Patrick Musicha [2,3,5], Mathew A. Beale [3], Yohane Diness[2], Oscar Kanjerwa[2], Chifundo Salifu[2], Zefaniah Katuah[2,6], Patricia Duncan[7], John Nyangu[7], Andrew Mungu[7], Muonaouza Deleza[8], Lawrence Banda[8], Lumbani Makhaza[2], Nicola Elviss[9], Christopher P. Jewell[10], Gina Pinchbeck[1], Nicholas A. Feasey [2,5,11], Eric M. Fèvre [1,12,14] ✉ & Nicholas R. Thomson [3,13,14]

Diverse salmonellae have the potential to cause disease and may be carried asymptomatically within the intestine of many vertebrate species. The relative contribution of human, animal, and environmental hosts to the transmission of *Salmonella* is unknown within and between households in low-income settings, especially where humans and animals may live in close contact and sanitary infrastructure is often inadequate. Between November 2018 and December 2019, we isolated *Salmonella* spp. from thirty households in urban and rural locations in Malawi, sampling at three time points from the stool of humans, animals, and their household environment. Using whole genome sequencing and fine-resolution bioinformatic and phylogenetic analyses we found evidence of sharing of *Salmonella* species and strains between humans, animals and the environment, both within and between households. The intricate web of interconnected salmonellae within this ecosystem underscores the importance of adopting a multi-faceted 'One Health' strategy when considering control of *Salmonella* in low-intensity agricultural systems.

It is now well established that ~75% of emerging and re-emerging pathogens affecting humans worldwide are zoonotic, with the greatest disease burden affecting poorer and more marginalised populations in low- and middle-income countries[1–4]. The One Health concept recognises that the health of humans, domestic and wild animals and their wider environment are closely linked and interdependent[5]. Over the last decade, changes to global ecosystems, demographics, socio-cultural and economic factors have reinforced the interconnection between humans, animals and their environment and amplified the need for collaborative, multi-sectoral and transdisciplinary approaches to understand and optimise responses to these changes. This has been accompanied by substantial fluctuations in climate and ecosystem health, which are associated with extension of the ranges of non-endemic pathogens

[1]Institute of Infection, Veterinary and Ecological Sciences, University of Liverpool, Liverpool, UK. [2]Malawi Liverpool Wellcome Programme, Blantyre, Malawi. [3]Wellcome Sanger Institute, Hinxton, UK. [4]Department of Veterinary Medicine, University of Cambridge, Cambridge, UK. [5]Department of Clinical Sciences, Liverpool School of Tropical Medicine, Liverpool, UK. [6]Malawi Adventist University, Blantyre, Malawi. [7]Department of Animal Heath and Production, Ministry of Agriculture and Food Security, Blantyre, Malawi. [8]Lilongwe University of Agriculture and Natural Resources, Bunda, Lilongwe, Malawi. [9]UK Health Security Agency, London, UK. [10]School of Mathematical Sciences, Lancaster University, Lancaster, UK. [11]The School of Medicine, University of Andrews, St, Andrews, UK. [12]International Livestock Research Institute, Nairobi, Kenya. [13]Faculty of Infectious and Tropical Diseases, The London School of Hygiene and Tropical Medicine, London, UK. [14]These authors contributed equally: Eric M. Fèvre, Nicholas R. Thomson. ✉e-mail: cnw25@cam.ac.uk; eric.fevre@liverpool.ac.uk

such as dengue virus, West Nile virus and *Vibrio cholerae*[6–8]. In the future, pathogen spillover from animal to human populations and vice versa may occur more frequently, particularly in the face of increasing urbanisation[9–11]. In addition, antimicrobial resistance (AMR) has emerged as one of the leading public health threats of the twenty-first century[12]. In 2019 it was estimated that the highest proportion of the global burden of deaths owing directly to drug-resistant infections occurred in sub-Saharan Africa, a consequence of prevailing levels of poverty leading to inadequate investment in sanitation and healthcare infrastructure, a high overall burden of infectious diseases, poor regulation of antimicrobial use and lack of alternatives to effective antimicrobials[13].

*Salmonella* spp. are an ideal model to investigate bacterial flux in a One Health context since they include several globally relevant pathogens and can also be carried asymptomatically within the intestine of a wide variety of vertebrate species, including animals reared for meat and egg production. Further, salmonellae can exist stably within the environment at ambient conditions for long time periods[14,15]. Worldwide, *Salmonella* spp. are estimated to cause 78.7 million human cases of gastroenteritis annually, with 59,100 deaths and 4.1 million disability adjusted life-years[16]. This estimate is lower on the African continent, where forty-six percent of cases of illness caused by non-typhoidal *Salmonella* (NTS) are attributed to exposure through the foodborne pathway[17]. In sub-Saharan Africa, NTS has become one of the most common causes of invasive bacterial bloodstream infection in humans over the last 40 years[18–20]. The most recent estimate states that invasive NTS accounts for 29.5% of cases of bloodstream infections in Africa, carrying an average case-fatality rate of 20.6%[21,22]. Most invasive NTS disease in Malawi can be attributed to the specific strain *S*. Typhiumurium ST313, with *S*. Enteritidis ST11 the second most common strain reported[23–28].

Alterations in the demographics and socio-economic status of populations may lead to changes in animal husbandry and management practices, increase proximity to and contact with humans for both domestic and wild animals and consequently altered incidence of endemic zoonotic disease. Population growth and urbanisation are fuelling a significant increase in the demand for meat production within sub-Saharan Africa[29]. Within this sector animals may often be reared and slaughtered in conditions with little provision for biosecurity, particularly within low-intensity production systems around households[30]. Domestic, livestock and poultry animals belonging to the household are often kept within the household perimeter, and in some settings, humans and animals sleep under the same roof[31]. Loosely structured waste management systems for human and animal species within these environments offer the opportunity for environmental faecal contamination and consequent exposure of peri-domestic wildlife species such as geckos, wild birds and rodents to human and animal faecal residues within the household environment[32]. These factors increase the potential for transmission of *Salmonella* spp. and AMR determinants carried by these bacteria.

To investigate sharing of *Salmonella* spp. within the extended household, considering humans, animals and the household environment, we conducted a prospective longitudinal surveillance study of thirty households from a high-density urban and a rural setting in Malawi. We collected human and animal faecal samples as well as environmental samples from frequently contacted household surfaces. Using whole genome sequencing we determined the relationships between *Salmonella* isolates, finding evidence of extensive sharing of *Salmonella* spp. between humans, animals and the environment, both within and between households. The insights gained into the dynamics of *Salmonella* spp. dissemination will inform future considerations for enhanced biosecurity and surveillance within such settings. In addition, our results support a broader understanding of the risks and drivers of pathogen emergence across interfaces.

## Results

### Description of the participating households

Between 19th November 2018 and 16th December 2019 thirty households were recruited in two study sites in Malawi. Fifteen houses were recruited in Ndirande, an informal urban settlement in Blantyre, and a further fifteen in the rural area Chikwawa (Supplementary Fig. 1). Each household was visited three times, the second visit taking place ~2 months after the first visit (median interval between first and second visit 63 days, range 35–140 days), followed by a third visit roughly 6 months after the first visit (median interval between first and third visit 204 days, range 120–330 days). In total 411 stool samples were collected from 184 humans. For each household, at least one environmental sample (total $n = 646$, median per household $n = 7$, range per household 1–19) was collected from areas of high human-human, animal-animal, or human-animal contact such as dwelling surfaces inside the household and animal pens, food preparation and water storage areas, dirty clothing, beds and latrine areas. All households kept at least one livestock, domestic or poultry animal and we collected 1023 animal stool samples from a range of livestock (cattle, sheep, goats, pigs), domestic animals (dogs, cats and guinea pigs), poultry (chicken, ducks, doves, guinea fowl, jungle fowl) and peri-domestic wildlife (rodents, geckos and wild birds). In total, 2080 individual samples were collected, 965 (46.4%) from Ndirande and 1115 (53.6%) from Chikwawa. Samples were cultured and the presence of *Salmonella* was screened for by PCR for *ttr*. PCR for *ttr* was positive in 233/2080 samples (11.2%) (Supplementary Fig. 2, Supplementary Table 1–3). In total 87/965 (9.0%) of samples from Ndirande and 146/1115 (13.1%) of samples from Chikwawa were positive for *Salmonella* spp.

### Distribution of *Salmonella* genomes

Whole genome sequencing was used to confirm the presence of *Salmonella* spp. We performed DNA extraction and whole genome sequencing of two-three *ttr* positive colony picks per sample to enable us to assess multi-serovar carriage and within-host diversity. We therefore sequenced 403 isolates in total, which were taken from a total of 214 samples. After quality assessment of sequence data, 227 genomes were identified as *Salmonella* genomes, passing quality thresholds and were subsequently included in our analysis (Fig. 1a). These 227 genomes originated from 111 discrete samples.

On examination, this collection of 227 genomes contained a number of 'identical' bacterial isolates (i.e. identical bacterial isolates are detected within one sample, collected at the same household at the same timepoint). One isolate from a pair of identical isolates was removed (de-duplicated) from this collection, leaving a total of 131 individual isolates (an individual isolate is a single *Salmonella* isolate of each serovar or sequence type from each individual sample). These 131 individual isolates were collected from 111 samples, therefore more than one individual *Salmonella* isolate was identified from twenty of these samples (within-host diversity). The 111 samples originate from 81 animal stool samples (73.0%), 16 environment samples (14.4%) and 14 human stool samples (12.6%). At least one *Salmonella* genome was generated from 25/30 households in the study; 14/15 (93.3%) households sampled in Chikwawa and 11/15 (73.3%) households sampled in Ndirande (Fig. 1a).

In total 94/1115 (8.4%) of samples collected from Chikwawa and 17/965 (1.8%) of samples collected in Ndirande contained *Salmonella*. Of the 111 samples, 53/111 (47.7%) were collected from samples taken during the first visit to the household, 32/111 (28.8%) during the second visit and 26/111 (23.4%) during the third visit. *Salmonella* genomes were generated from 6/30 (20%) households at all three visits, 11/30 (36.7%) households at two visits, 8/30 (26.7%) households at one visit. Considering human, animal and environmental samples as separate 'compartments', *Salmonella* genomes were

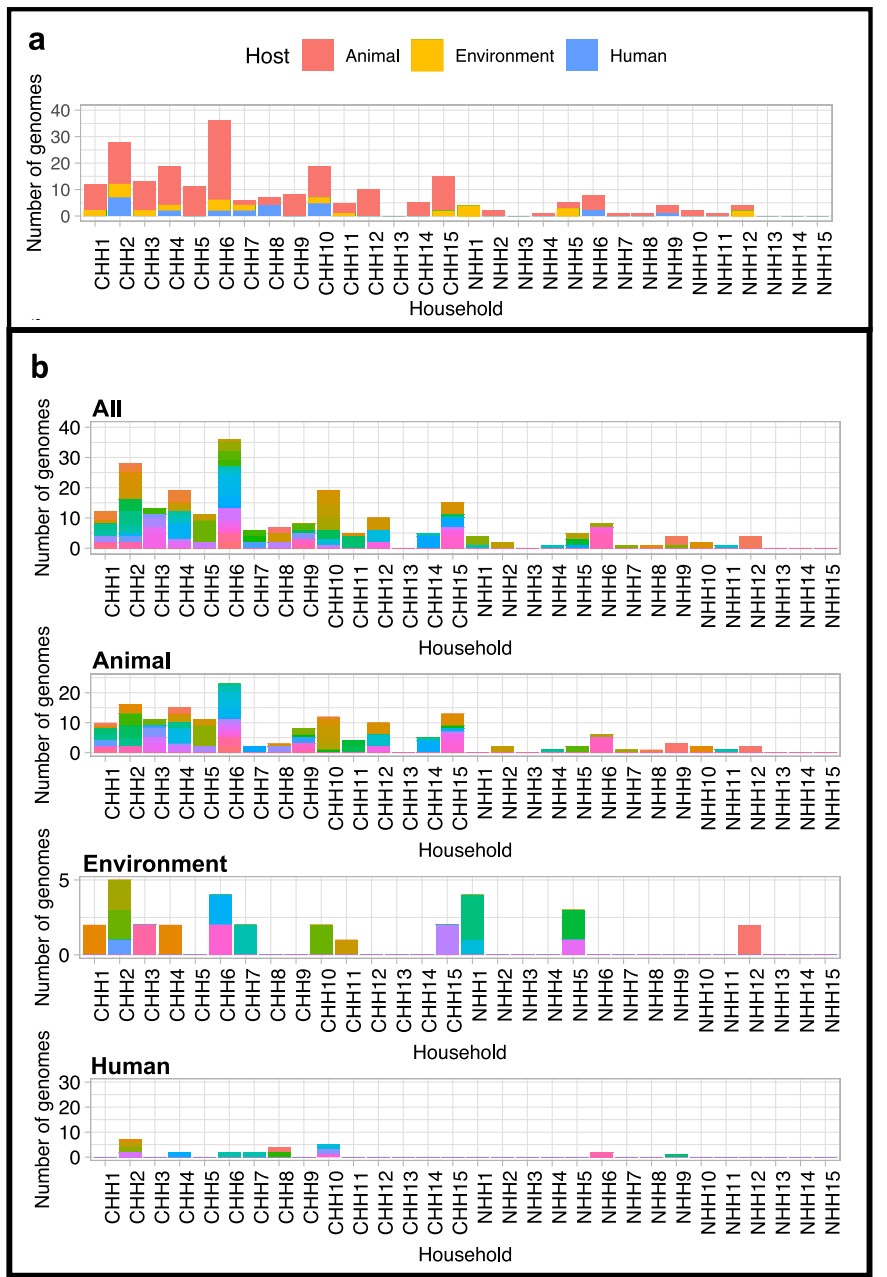

**Fig. 1 | Number and diversity of *Salmonella* genomes isolated from each household. a** Total number of good quality genomes isolated from human, animal hosts or the environment at each household during the study (*n* = 227). **b** Diversity of *Salmonella* sequence types isolated from each household and each host category (human, animal or the environment) within the collection of 227 genomes. Coloured bars represent each individual sequence type of which 64 were detected within the collection of 227 genomes. CHH = household located in Chikwawa study site, NHH = household located in Ndirande study site followed by an identifying number of the household. Source data are provided as a Source data file.

generated from all three compartments at 5/30 (16.7%) households, two compartments at 9/30 (30%) households, and one compartment at 11/30 (36.7%) households.

We used the real-time analytic and genomic epidemiology platform Pathogenwatch[33] to delineate *Salmonella* genomes into species, subspecies and serovars. Isolates in the collection were drawn from two subspecies of *Salmonella enterica*, comprising 125 (55.1%) isolates of *Salmonella enterica* subspp. *enterica* (*S. enterica*), and 102 (44.9%) isolates of *Salmonella enterica* subspp. *salamae* (*S. salamae*) (Fig. 2). Among these subspecies, 56 serovars were identified within the collection, 26 *S. enterica* and 30 *S. salamae* (Supplementary Fig. 3).

The 227 *Salmonella* genomes were identified from human stool (*n* = 25, 11.0%), animal stool (*n* = 171, 75.3%) and environmental

samples (*n* = 31, 13.7%)(Fig. 2). There was no significant difference (chi squared test, *ρ* = 0.49) between the number of *S. enterica* and *S. salamae* genomes collected from each host category. *Salmonella* was detected within the stool of twelve different animal species (chicken, duck, dove, guinea fowl, gecko, wild bird, rodent, cockroach, cattle, pig, goat, dog).

A surprising number of *S. salamae* genomes were detected within the study, offering a chance to investigate this little documented subspecies of *Salmonella*. Of the thirty-two *Salmonella* genomes detected in total from Ndirande, twenty-one (65.6%) were *S. enterica* and eleven were *S. salamae* (34.4%). Of the 195 genomes in total from Chikwawa, 104 were *S. enterica* (53.3%) and 91 were *S. salamae* (46.7%). There was no significant difference between the number of *S. enterica*

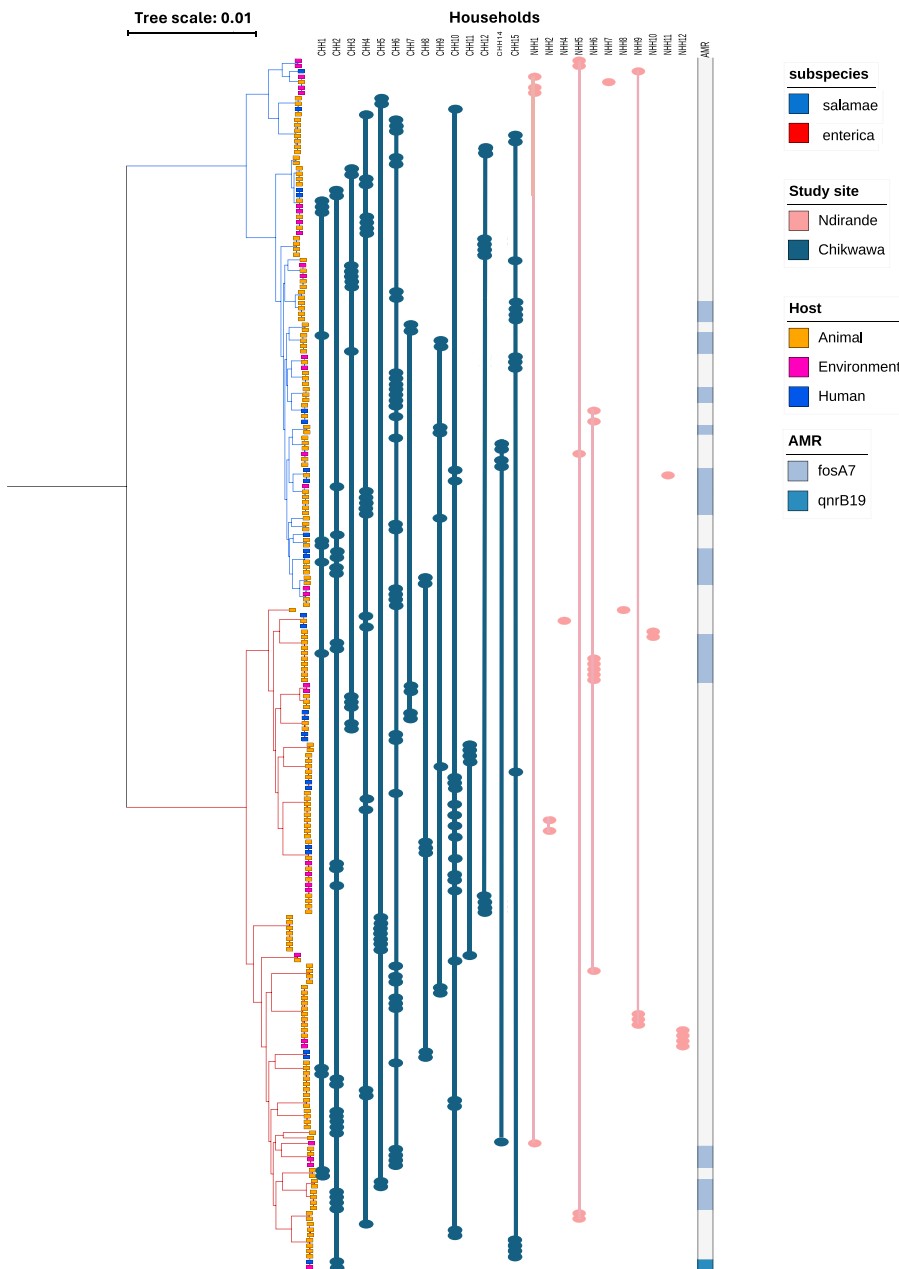

**Fig. 2 | Phylogenetic tree of the 227 genomes within the collection to show the diversity of *Salmonella* present within each household and amongst the different hosts sampled.** The different coloured tree branches represent the two different *Salmonella* subspecies present in the collection. The tips of the tree are coloured according to the host species from which the genome was isolated. The household from which each *Salmonella* genome was isolated (identified at the top of the figure) is depicted by a filled-circle, coloured according to study site.

Coloured bars linking the *Salmonella* genomes (filled-circles), also coloured by study site, join salmonellae identified within a single household together. Genotypic antimicrobial resistance determinants detected displayed in heatmap on the right of the figure. Shades of blue depict the presence of an AMR determinant, white depicts no AMR determinant detected within the genome. *fosA7* and *fosA7.7* have been visualised together as *fosA7*. *gyrB*, as a nonsynonymous mutation, is not visualised here. Source data are provided as a Source data file.

and *S. salamae* genomes detected at each study site (chi-squared test, $\rho = 0.15$) demonstrating that the two subspecies of *Salmonella* were equally represented across both sites.

### Inferring the distribution of plasmids, antimicrobial resistance determinants and virulence genes within the collection

Multidrug resistance is an important concern for the treatment of invasive and noninvasive NTS disease in humans and animals[34,35]. Given the previously documented increase in AMR in *Salmonella* in this

setting[36], we investigated the distribution of AMR determinants, plasmids and virulence genes. MOB-suite[37] was used to identify a total of 85 plasmid replicons in 72/227 genomes. Eight different plasmid replicon types were indentified across 61 plasmids, of which IncFII was the most commonly identified (20/227, 8.8% genomes) followed by IncFIB (16/227, 7.0% genomes). In 12 genomes, we found two co-occurring plasmid replicon types, and for a single genome we found three co-occurring plasmid replicons, all located on different contigs. For the remaining 24/85 (28.2%) putative plasmids, we were unable to assign a

plasmid replicon type; 20 of these were detected within *S. salamae* isolates (Supplementary Fig. 4).

We used abricate[38] to infer the presence of known virulence genes, identifying 126 different virulence genes within the collection, 40 (32%) of which were uniformly present in all 227 genomes. Thirty-one virulence genes (25%) were only detected within *S. enterica*, of which most were predicted to encode Type 3 Secretion System proteins. Virulence genes were carried by 23/85 plasmids (27.1%). None of these plasmids were conjugative, 11/23 (47.8%) were mobilisable (require a helper plasmid for conjugation to occur). Importantly, given the potential for horizontal transmission of plasmids between salmonellae, in this collection none of the plasmids which carried virulence genes also carried AMR determinants.

We found four different types of AMR determinants across 47 genomes (20.7%) (*fosA7*, *fosA7.7*, *qnrB19* and *gyrB19* non-synonymous mutation). *fosA7* and *fosA7.7* confer resistance to fosfomycin (not confirmed phenotypically here), while *qnrB19* confers resistance to quinolones (and low level fluoroquinolone resistance, confirmed phenotypically for 3/4 genomes carrying *qnrB19*). No genomes contained more than one AMR determinant. The most common AMR determinant was *fosA7*, detected in 36/227 genomes (Supplementary Fig. 4). Both *fosA7* and *fosA7.7* were located on chromosomal contigs. Four *qnrB19* genes were identified on plasmid contigs, all identified as plasmid rep cluster 2355, within *S.* Typhimurium ST19. These four isolates demonstrated phenotypic resistance to nalidixic acid. There was a single nonsynonymous SNP mutation of *gyrB* (S464F) detected in each of four genomes, all present on chromosomal contigs of *S.* Enteritidis. These four isolates carrying the *gyrB* mutation were associated with intermediate phenotypic resistance to nalidixic acid.

## Genetic diversity of salmonellae genomes

To assess the genetic diversity of the isolates in our collection, we determined multi-locus sequence types (MLST) for all *Salmonella* genomes, finding 64 different *Salmonella* sequence types (STs) across the two subspecies (Fig. 1b). Of these, only 6 STs (9.4% of genomes) were present in both Chikwawa and Ndirande, indicating a lack of sharing between the two sites, whilst 50 STs (78.1% of genomes) were present only in rural Chikwawa, and 8 STs (12.5% of genomes) were present only in urban Ndirande (Fig. 1). Across both sites, the median number of STs detected within each household was three, but this was higher for Chikwawa (*n* = 5, range 0–15) than for Ndirande (*n* = 1, range 0–3). The median number of *S. enterica* STs collected from each household in Chikwawa was 2 (range 0–7), that of *S. enterica* in Ndirande was 1 (range 0–2). By contrast, the median number of *S. salamae* STs in the whole collection was 1 ST from each household, the median number of *S. salamae* isolates collected from households in Chikwawa was 2 (range 0–9) and the median number of *S. salamae* isolates collected from households in Ndirande was zero (range 0–2).

## Phylogenomic distribution of salmonellae

We used Panaroo[39] to infer a pangenome consisting of 11,964 unique genes (coding sequences), of which 3429 genes were defined as core (present in 98% of genomes), representing 28.7% of the pangenome, with the remaining 8535 (71.3%) genes forming the accessory genome. A core gene phylogeny (Fig. 2) was inferred to determine genomic relatedness of genomes collected within and between households. Pairwise SNP distances were calculated using the core gene alignment of the entire collection of 227 genomes (total alignment length 3,065,105 bps, 241,674 (7.9%) variable positions), and additionally using two species-specific core gene alignments for 125 *S. enterica* subsp. *enterica* genomes (total alignment length 3,240,911, 133,718 (4.1%) variable positions) and 102 *S. enterica* subsp. *salamae* genomes (total alignment length 3,239,471, 100,418 (3.1%) variable positions) (Supplementary Figs. 5 and 6). Across our entire collection of 227 genomes, 1841/25,651 (7.2%) sample pairs originated from within the

same household and 23,810/25,651 (92.8%) sample pairs were from different households (Table 1). The median number of pairwise SNPs between any two samples amongst the whole collection was 36,847 SNPs using the core gene alignment of 227 genomes. There was a significant difference between the number of SNPs between pairs of genomes detected within the same household (median pairwise SNP distance = 34,931 SNPs), compared to the number of SNPs between pairs of genomes originating from different households (median pairwise SNP distances was 86,956 SNPs; Kruskal Wallis test, *p* < 0.001).

We examined the pairwise SNP distances present within each *Salmonella* subspecies amongst samples originating from the same or different households (Table 1) and observed an extremely high degree of diversity within the collection as a whole. We also found a small percentage of identical genomes with 0 pairwise SNPs using the core gene alignment (Table 1).

## Sharing of *Salmonella* between hosts

We used subspecies-specific (*S. enterica* and *S. salamae*) pairwise SNP distances to investigate whether genetically related *Salmonella* were present within two or more epidemiologically linked hosts within the study. Based on the relatively large number of close genomic relationships between samples from the same household described above (Table 1) we selected a conservative approach, considering genomes separated by zero pairwise SNPs isolated from different hosts and sources to be putative 'shared' pairs, either by direct transmission or recent acquisition from a common source. As described earlier, the final collection of 227 genomes consist of a collection of 131 individial isolates collected from a total of 111 samples, as two-three colony picks were submitted from each sample to assess within host diversity. In order to investigate putative sharing of genomes between hosts and to minimise bias owing to within-sample repeat sampling, we deduplicated identical (0 SNPs) pairwise SNP distance measurements of *Salmonella* genomes from the same individual sample taken at the same household at the same visit to the household (Supplementary Figs. 5 and 6).

This left 20 pairs of genomes sampled from different hosts with a pairwise SNP distance of 0 SNPs (Fig. 3). Ten of these genome pairs were *S. enterica* and ten *S. salamae*. This represented 0.12% (0.07–0.2%) of the total number of *S. enterica* genome pairs (*n* = 7750 pairs) and 0.19% (0.11–0.36%) of the total number of *S. salamae* genome pairs (*n* = 5151 pairs). Eleven (11/20, 55%) of these genome pairs occurred within household and 9 (9/20, 45%) occurred between households (Table 2, Supplementary Figs. 5 and 6).

No putative sharing pairs were found solely within our urban Ndirande site, 16 putative sharing pairs were found within rural Chikwawa and 4 putative sharing pairs were identified between Chikwawa and Ndirande. This is interesting as the two study sites are 50 km apart and we found no epidemiological connections between either human or animal hosts between each site.

In our dataset, 11/20 (55%) putative sharing events were from animal-animal host pairs. In contrast, we did not find any human-human host pairs, nor any environment-environment host pairs (Table 2, Fig. 3). Of our putative animal-animal sharing pairs, 5/11 (45%) occurred within the same animal species, whilst 6/11 (55%) occurred between animal species (Fig. 3).

There were seven pairs of *Salmonella* genomes (7/20, 55%) which were shared between humans and either an animal or the environment (Fig. 3). Two of these were within the same household, one isolate of each pair collected at separate visits to the household. For example one *Salmonella* sample collected from the swab of an outside tap during the second visit to the household in Chikwawa was 0 SNPs different to *Salmonella* detected within the stool of a boy living at the same household, collected during the third visit to the household 6 months later. A second sharing pair involved *S.* Johannesburg from an adult human male collected during the first visit to the household

**Table 1 | Description of the number of genome pairs and pairwise SNP distance in the total collection and within and between household pairs of genomes (Supplementary Figs. 5 and 6)**

| | Whole collection (n = 227) | | | Sub.sp enterica (n = 125) | | | Sub.sp salamae (n = 102) | | |
|---|---|---|---|---|---|---|---|---|---|
| | Total | Between household | Within household | Total | Between household | Within household | Total | Between household | Within household |
| Number of genome pairs | 25,651 | 23,810 | 1841 | 7750 | 7200 | 550 | 5151 | 4688 | 463 |
| Median SNP distance (SNPs) | 36,847 | 86,956 | 34,931 | 34,638 | 34,820 | 25,563 | 15,845 | 15,929 | 13,211 |
| Range (SNPs) | 0–88,886 | 0–88,886 | 0–88,667 | 0–38,198 | 0–38,198 | 0–37,922 | 0–38,712 | 0–38,712 | 0–37,967 |
| Closely related (<or equal to 100SNPs) | 414 | 207 (50.0) | 207 (50.0) | 275 | 150 (54.9) | 123 (45.1) | 140 | 56 (40.0) | 84 (60.0) |
| Very closely related (<or equal to 10SNPs) | 170 | 23 (13.5) | 147 (86.5) | 110 | 15 (13.6) | 95 (86.4) | 91 | 15 (16.5) | 76 (83.5) |
| 'Identical' genome pairs (0SNPs) | 67 | 8 (11.9) | 59 (88.1) | 51 | 4 (7.8) | 47 (92.2) | 26 | 7 (26.9) | 19 (73.1) |

Whole genome collection results are gained using the core gene alignment of 227 genomes. S. enterica results are gained from a core gene alignment of 125 S. enterica genomes only, and S. salamae results are gained from a core gene alignment of 102 S. salamae genomes only. In brackets are shown the percentage of the whole genome collection, S. enterica and S. salamae genomes which are either closely related, very closely related or 'identical' which were generated from samples collected from within the same household or from two samples collected from different households.

and a chicken sampled at the third visit to the same household 6 months later. Both households were located within Chikwawa and were livestock and poultry-owning households in which the animals (domestic animals, livestock and peri-domestic wildlife) and humans shared the same living space within the household. The remaining five pairs were shared between human and animal samples taken at different households. There was no sharing detected between wild birds and other hosts within the study.

Within the collection of 227 genomes only four pairs of isolates with 0 pairwise SNPs in the core gene alignment shared the same AMR determinants. Three pairs of *S. salamae* genomes shared *fosA7*, a genomic AMR determinant which confers phenotypic resistance to the antibiotic fosfoycin. *fosA7* was chromosomally integrated within these *S. salamae* genomes. We detected one pair of *S.* Typhimurium ST19 genomes which shared the determinant *qnrB19* (Fig. 3b) which confers phenotypic quinolone resistance and low level fluoroquinolone resistance. The *qnrB19* AMR genes were detected within plasmids classified by MOB-suite as replicon cluster 2355. These plasmids are non-conjugative and require a 'helper' plasmid possessing the necessary relaxase and Mpf genes to be mobilised. Three of these pairs were detected within the same household, one between households.

## Discussion

In this study, we show that closely related salmonellae are shared between humans, animals and the environment both within and between households in Malawi, particularly in rural areas. In these rural areas, humans are often involved in low-intensity agricultural practises, with the household itself a base in which both humans and animals reside overnight. This means that humans and animals of a variety of species often spend at least part of each day within close proximity. *Salmonella* has provided an excellent model to document the occurrence of bacterial sharing around these household sites.

The study setting in this work differs markedly from those in other areas, particularly more industrialised agricultural settings, where the interaction between humans, animals and their shared environment may be more limited[40–43]. Previous work has looked at strain- and resistome-sharing of *Escherichia coli* in sub-Saharan Africa between humans, animals and the environment in households in the urban setting of Nairobi, and demonstrated that sharing does occur between different host populations[44]. This study investigates sharing of *Salmonella* within households located in both rural and urban environments in sub-Saharan Africa. We collected samples from a diverse range of animal species that spend time within the household perimeter in which the humans reside and we used a detailed longitudinal household sampling strategy to investigate distribution and dissemination of *Salmonella* between humans, animals and the environment. This sampling framework enabled us to correlate epidemiological linkage occurring at the household level with genomic relatedness of strains.

Within the study, a diverse collection of *Salmonella* genomes spanning two subspecies was detected. These isolates were carried apparently asymptomatically in the stool of a range of hosts, or detected within the environment. Importantly, whilst asymptomatically carried amongst humans and animals from Malawian households within this study, many of the serovars have been previously reported to cause clinical disease in other settings[25,26,45–49]. The connection and relationship between carriage and disease of NTS is not yet fully clear across all serovars, in all species. Encouragingly, we also documented low rates of AMR determinant carriage, and no multidrug resistance was detected within isolates collected within these low-intensity agricultural settings. However, we are aware that there is generally little regulation of antimicrobial use both discretely for animals and as an additive component in animal feed in settings such as these, which may drive the spread of AMR in the future, and should be monitored[50].

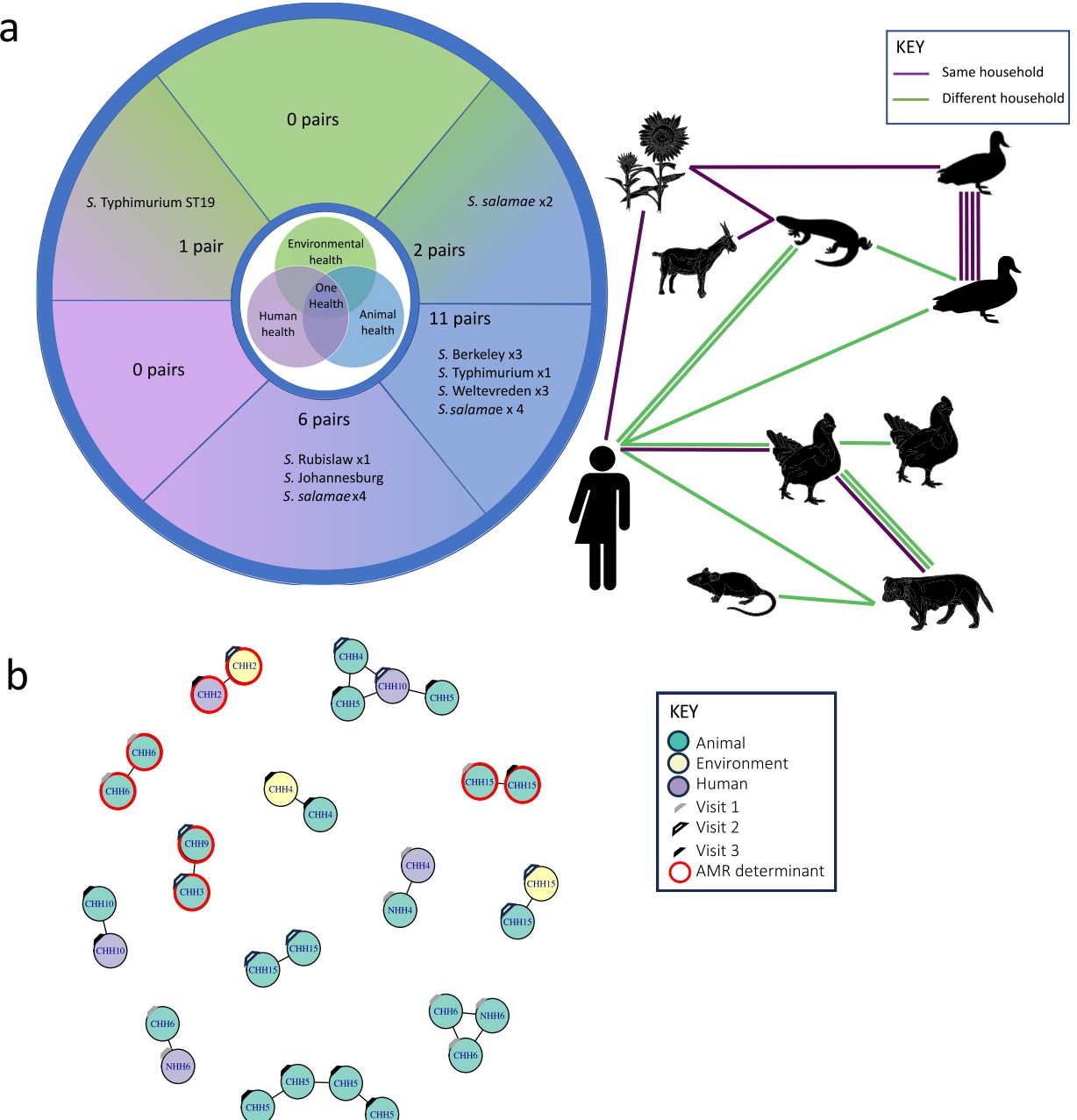

**Fig. 3 | Sharing pairs of *Salmonella* detected within the study. a** A composite map to show the nature of sharing pairs of *Salmonella* detected within the study within a One Health perspective. Human silhouette indicates genomes initially collected from humans. Animal silhouettes represent *Salmonella* initially collected from animals. The tree silhouette represents salmonellae collected from within the environment of each household perimeter. Each silhouette represents the host category or animal species as a whole. Number of lines between silhouettes indicates the number of sharing events noted within the collection. The colour of the line between silhouettes denotes whether the sharing event occurred between isolates of the same household or different households. Sections of the circle correspond to human, animal, environment pairs as labelled by the central graphic or human-environment, human-animal, animal-environment between the corresponding sections. Illustration from NIAID NIH BioArt Source[89–96] (Lizard Outline: Bioart.Niaid.Nih.Gov/Bioart/302; Sunflower: Bioart.Niaid.Nih.Gov/Bioart/620; Goat: Bioart.Niaid.Nih.Gov/Bioart/636; Duck Silhouette: Bioart.Niaid.Nih.Gov/Bioart/135; Domestic Chicken: Bioart.Niaid.Nih.Gov/Bioart/131; Domestic Dog: Bioart.Niaid.Nih.Gov/Bioart/594; Lab Mouse: Bioart.Niaid.Nih.Gov/Bioart/279; Unisex Icon: Bioart.Niaid.Nih.Gov/Bioart/13) **b** Network produced using iGraph to show the occurrence of pairs of *Salmonella* with a SNP distance from the core genome alignment of 0 SNPs. The colour of the node denotes host species (green = animal, yellow = human, purple = animal/environment). The household number from which the isolate was sampled is shown in text in the centre of each node. CHH = Chikwawa, NHH = Ndirande followed by an identifying number of the household. A red perimeter of the node indicates that the isolate carries one antimicrobial resistant determinant. In all cases the AMR determinant is shared between the pair. Colour of mark on upper right quadrant of circle denotes visit number. Source data are provided as a Source data file.

Consideration should be given to the location within households in which sharing of *Salmonella* occurs in Malawi. Animals pass faeces freely in the compound and frequently share the same environment in which children play and food is prepared. Risk factors for the carriage of *Salmonella* within human stool in Malawi have been found to be linked to animal ownership and husbandry factors and so it is important that improved environmental hygiene within the household should be addressed in the context of sharing of *Salmonella*[31].

**Table 2 | Description of the pairwise SNP distance of pairs of salmonellae of 0 SNP of the within-*Salmonella* subspecies core gene distance (deduplicated samples only)**

| Category | *salamae* (*n* = 10) | *enterica* (*n* = 10) | Total (*n* = 20) | Total number of pairs within collection | Proportion of 0 SNP pairs within collection % (95% CI) |
|---|---|---|---|---|---|
| Sharing within or between household | | | | | |
| Within household | 4 | 7 | 11 | 2026 | 0.54 (0.3–0.97) |
| Between households | 6 | 3 | 9 | 23,776 | 0.04 (0.02–0.07) |
| Sharing between hosts | | | | | |
| Animal-Animal | 4 | 7 | 11 | 7357 | 0.15 (0.08–0.27) |
| Human-human | 0 | 0 | 0 | 146 | 0 |
| Environment-Environment | 0 | 0 | 0 | 227 | 0 |
| Animal-Environment | 2 | 0 | 2 | 2616 | 0.08 |
| Human-Environment | 0 | 1 | 1 | 383 | 0.26 |
| Human-animal | 4 | 2 | 6 | 2172 | 0.28 (0.13–0.60) |
| Sharing between study sites | | | | | |
| Chikwawa | 9 | 7 | 16 | 9348 | 0.17 (0.11–0.28) |
| Ndirande | 0 | 0 | 0 | 286 | 0 |
| Chikwawa-Ndirande | 1 | 3 | 4 | 3267 | 0.12 |

Confidence intervals displayed in brackets where there are more than five pairs within each category.

Within this collection of diverse salmonellae we only very rarely detected strains of *Salmonella* which have been previously found to be associated with invasive NTS disease in Malawi[25,27]. We detected one *S.* Typhimurium ST313 isolate collected from the stool of a dog within one household at one timepoint, and isolates of *S.* Enteritidis ST11 have been detected within stool of dogs and the environment of two households. Further investigation of the relationship of these potentially invasive strains with previously published isolates demonstrates that the ST313 is more closely related to *S.* Typhimurium ST313 Lineage 3, which has been found to have emerged in Malawi in 2016, and the *S.* Enteritidis ST11 is closely related to those isolates of the outlier cluster which have been responsible for significant human disease in other settings (Supplementary Figs. 7 and 8)[25,27]. Within this study we have detected strong epidemiological links between *Salmonella* of relevance to human disease, but not to pathovars strongly associated with iNTS in Africa. This is predominantly consistent with previous findings, however, the potential for foodborne transmission cannot be discounted, as *S.* Enteritidis ST11 of the global epidemic clade has been previously isolated from samples collected within the livestock and poultry meat pathway in Tanzania[51–53]. This may be a consequence of a small sample size, but does also lead to the question of what are the reservoirs and key transmission routes of these pathovars?

Despite evidence of of *Salmonella* circulation within and between households across the two study sites, we did not document social connection between any of the households within the study. We have, however, sampled extremely sparsely; peri-domestic wildlife or free-roaming domestic or livestock animals may pass between and amongst households within a single study site, as described elsewhere, but equally we lacked the resolution to confirm an epidemiological link[54]. We suggest that *Salmonella* has been resident for a long period of time across both study sites, and recent spread of these particular genomes which are present within both study sites, has occurred. Considering the mutation rate of *S.* Typhimurium (6.7 SNPs per genome per year) it is reasonable to assume that this recent spread has occurred over the last 10 years[55].

The total number of genome pairs detected within this study is low and therefore it is not possible to further quantify the sharing which has occurred. However, given the small number genomes (*n* = 227) overall, and in Ndirande (*n* = 33) specifically, amongst which sharing was investigated, it may have been expected that sharing could have been missed entirely. Therefore, investigation of a larger study population is warranted, in order to further quantify the sharing of *Salmonella* within and between households, study sites and hosts.

The most common type of sharing was between chicken and dogs. The connection of *Salmonella* sharing between dog and poultry is interesting. In Chikwawa dogs and poultry are often free-roaming around households during the daytime, and poultry are penned at night. As omnivores and scavengers, dogs, ducks and chickens have access to food and faeces which may be shared around the ground of the household, providing plenty of opportunity for faecal-oral transmission to occur. It may be that the presence of these animals within a household acts as an ecological driver for the sharing of *Salmonella* between hosts. A range of environmental health practises have been shown to be important to reduce the sharing of bacteria and AMR determinants[31,56,57]. Implementation of more stringent biosecurity procedures specifically as part of animal husbandry practises, including regular removal of animal faeces from around the household complex and improved hand hygiene are also profoundly important in the endeavour to reduce the potential spread of these bacteria within households in Malawi, and should be implemented alongside interventions.

The majority of the world's population live in developing economies and small, low-intensity farms produce ~35% of the world's food[58,59]. We show that sharing of identical salmonellae between humans, animals and the environment is possible and in fact likely, demonstrating the importance of considering all aspects of hygiene and biosecurity precautions within households when developing strategies to limit the movement, carriage and sharing of salmonellae and other gastrointestinal pathogens. The findings of this study have important implications for public health, livestock keeping and animal husbandry policy and practice in low-income, low-intensity farming settings and should be used to shape efforts to draft effective, durable evidence-based policies to safeguard human health and ensure sustainable livestock systems in these settings.

## Methods
### Study site
Between November 2018-December 2019 a longitudinal prospective study recruited 30 households from two study sites in Malawi: Ndirande, Blantyre District and Chikwawa District. Within each geographic area polygons were created using QGIS software to create areas for inclusion[61]. Fifteen households were selected at random in the two study areas using R software version 2022.12.0 + 353 to generate

random GPS coordinates using a spatial inhibitory design with close pairs[62]. Households in each location met the inclusion criteria of being located within the study sites, all human household members were able to give informed consent or assent to take part in the study themselves, and the Head of the Household was able to provide informed consent to sample animals and the environment within the household. Households were excluded if a household member or representative was unable to provide informed consent, household members spoke neither Chichewa or English and if the household was located outside the boundary of the study sites. Sampling was carried out at three time points in all households. Identical sampling procedures were used at each time point to collect samples from humans, animals and the environment.

## Sample collection

Questionnaires detailing household composition, socioeconomic data, animal ownership, husbandry, contact of members with animals, health seeking behaviour of humans were administered at each household using an electronic case report form on a Samsung© tablet device using Open Data Kit Collect version 1.18[63]. At each visit to a household, faecal samples were collected from all consenting human participants and stool samples, rectal or cloacal swabs were taken from a representative number of each species of animal and/or birds present in the household. Environmental samples were collected using 3 M® swabs from areas of suspected high human-human, animal-human, animal-animal contact.

## Method of faecal sample collection

The method of faecal sample collection is explained in detail below.

**Human samples.** Fieldworkers left sterile faecal sample containers for each human study participant in the household on Day 1 of sampling. These were clearly labelled to identify which container should be used for each participant. Participants were also given nitrile gloves, biodegradable bowls and sample containers with spoons to facilitate collection of the sample. The bowls could be disposed of in a pit latrine or collected by the fieldworkers for hygienic disposal at the same time as stool sample collection. Families were given sealable opaque 'freezer bags' or similar to store samples whilst waiting for collection. Samples were collected on Day 2.

**Domestic animal and livestock samples (dogs, cats, sheep, goats, cattle, pigs.).** On Day 2 faeces (2–20 g) were collected directly from the rectum of animals should they be available at the household and placed into appropriately labelled individual sample containers. Appropriate Personal Protective Equipment including wellington boots, a boiler suit and protective gloves were used for sample collection. Animals were appropriately restrained by trained personnel during sample collection. Collection per rectum was conducted either manually or using a sterile swab, depending on the size of the animal.

**Poultry (chickens, ducks, geese, guinea fowl, turkeys, jungle fowl, pigeons and doves.).** Domestic poultry were sampled on the morning of Day 2 prior to release from the overnight housing. Poultry were appropriately restrained and a single cloacal swab was obtained from each bird sampled by either an Assistant Veterinary Officer or the author.

## Peri-domestic wildlife samples

**Rodents.** Household members were supplied with a pair of nitrile gloves and an appropriately labelled sterile faecal sample container with which to do this. The faecal sample container was then collected at the same time as the human stool samples on the second day of sampling.

**Geckos.** Household members were taught how to recognise and collect gecko excreta on Day 1 of sampling at each household and appropriate sterile faecal sample containers and nitrile gloves were left with the household in order that the gecko faeces be collected overnight prior to collection of the pot on Day 2. If household members had not collected gecko faeces, the field team would perform the sample collection on Day 2.

**Wild birds.** Again, following a pilot study to trial the efficacy of using bird nets to collect samples, the preferred method for wild bird faecal sample collection was found to involve placing appropriate clean tarpaulins underneath roosts located within the household perimeter on the morning of Day 2, upon first arrival at the household. Any faeces deposited on the tarpaulins by wild birds were collected and pooled into an appropriately labelled sterile sample container at the end of sampling on Day 2.

## Environmental samples

Environmental samples were collected individually using sterile 3 M® swabs. Each 3 M® swab contains a sterile sponge swab and 10 ml sterile buffered peptone water. Each sample was taken in a sterile manner, repeatedly rubbing a fresh 3 M® swab over an area of the object to be sampled of up to 20 × 20 cm for 30 s. The soiled swab was then replaced directly into the original sterile 3 M® swab packet, the plastic handle broken off and the packet securely fastened. The nature of location or object from which the swab was taken was recorded on the outside of each packet.

Where possible, environmental swabs were taken from certain consistent areas at each household and at each visit. These areas which included the door or curtain to the latrine, around the edge of the latrine, cooking areas, front door to house, the inside of water carriers, bootsocks on the floor outside the house, bootsocks on the floor inside the house, a dirty chitenje (ladies' skirt-wrap), surface of bed.

Bootsock samples were collected using plastic overshoes. Whilst wearing clean nitrile gloves, two clean plastic overshoes were placed onto one foot of the fieldworker or author. The fieldworker or author walked around the area of interest for roughly 1 min. Once the bootsock sample had been collected, whilst wearing clean nitrile gloves, the outer plastic shoe cover was removed and placed into a sterile self-sealing, appropriately labelled, plastic bag, sealed and placed into the cool box for transportation to the laboratory.

The total number of household environmental samples taken per visit was normally up to one third of the total samples taken, dependent on the total number of animal species present and total number of humans providing samples.

Samples were stored at 4 °C with ice packs in a cool box until arrival at the College of Medicine (now KUHES) laboratories. Laboratory processing commenced within 4 h of sample collection.

## Follow-up sampling

Samples were taken from each of the fifteen randomly selected households at three time points:

- TP0
- TP1 = TP0 + 2 months
- TP2 = TP0 + 6 months

Aside from the initial consenting steps, identical human and animal sampling strategies were used at each time point.

## Microbiological testing

Salmonellae were isolated and identified by selective culture using enrichment steps using buffered peptone water and Rappaport Vassiliadis for 24 h each respectively and a loop of bacterial solution from each were streaked out onto each of CASE and XLD selective agar

plates prior to incubation at 37 °C (Supplementary Fig. 2). From the XLD and CASE agar culture plates up to 5 colonies per sample were randomly selected and underwent O and Vi antigen testing to confirm the presence of salmonellae, and the absence of typhoidal-*Salmonella*. Once isolated, aliquots of pure bacterial growth were stored at −80 °C in individual microbanks or modified microbank tubes. Up to five picks of suspected *Salmonella* isolates from each sample were stored.

## *Salmonella ttr* qPCR

Quantitative PCR using bacteria extracted using the boilate method was carried out of all stored colonies of suspected *Salmonella* at the Malawi Liverpool Wellcome Programme[64]. Positive qPCR confirmation of each isolate was denoted by the presence of the tetrathionate reductase *(ttr)* gene[65]. This gene is involved in the respiration of *Salmonella* and is constitutively expressed in all salmonellae. The assays were run on a QuantStudio 7500 PCR machine. All qPCR assays in this study were run for 40 cycles. The highest acceptable cycle threshold is 35, but maintaining the cycle threshold at 40 allows for the identification of late amplification of the DNA (without it being due to chimeras). A standard curve (serial amplification of amplification target for which the concentration is known) was included in each run to allow estimation of the *Salmonella* bacterial load (copy numbers).

## Primers, Master mix and probe

The reaction mix is detailed in Supplementary Table 1. The primers were chosen to detect presence of the tetrathionate reductase (*ttr*) gene which is constitutively present in all *Salmonella* spp.[65]. The primers used were *ttr*−4 (AGCTCAGACCAAAAGTGACCATC) and *ttr*−6 (CTCACCAGGAGATTACAACATGG) (Supplementary Table 2), made up and supplied by Sigma[66]. Primers, probe and Master mix were stored at −20 °C.

## Controls

Three control samples were used in the PCR reaction. A positive control (*S.* Typhimurium NCTC), a negative boilate control (sterile distilled water only) and a negative PCR control (Master Mix omitted, other constituents were present). Results were analysed by reviewing positive and negative controls, adjusting the cycle threshold for detection above any background noise and reviewing the standard curve. The cycle threshold was set at the beginning of the exponential curve in the linear graph, and the middle of the linear phase of the log graph. For the standard curve a correlation coefficient ($R^2$) of >0.9, amplification efficiency of >80% and a minimum of 5 points within the assay linear range was considered adequate. Cycle threshold vales of the standard curve were also checked against typical and expected values.

For the run to be accepted all negative controls had to be below the threshold with no amplification and positive controls had to demonstrate a cycle threshold value <35 and a sigmoid curve. Analysis was performed by the laboratory technicians, reviewed and approved by the author. Assays were repeated when samples failed quality control.

## Reaction procedure

Following preparation of the reaction mix the procedure was as follows.

1. 22.5 μl of Master Mix, primer, probe and nuclease free water solution (Supplementary Table 1) were loaded into each well to be used of a new 96-well fast optical plate.
2. 2.5 μl of sample were added to the appropriate wells, including the controls.
3. Optical seals were applied to seal the plate.
4. The plate was spun for 5 s in plate centrifuge.

5. Plate cycled at the following temperatures for each reaction, for 40 cycles:
   a. Denaturation 30 s 95 °C
   b. Annealing 30 s 60 °C
   c. Extension 10 s 72 °C

The ramping up and down of temperature was set to 1.6 °C per second.

## Outcome and DNA extraction techniques

Isolates which were positive for the *ttr* gene were deemed to be *Salmonella* (previous work had confirmed that these were not typhoidal-salmonellae). These *Salmonella* isolates were stored and DNA extraction subsequently carried out using Qiagen DNA extraction kits (Qiagen DNA Mini kit). The DNA of two-three frozen colonies per sample was quantified as required by the guidelines of the Wellcome Sanger Institute using a Qubit© (Thermo Fisher Scientific, MA, USA). The extracted DNA was then sent for whole genome sequencing at the Wellcome Sanger Institute, UK.

## Whole genome sequencing and bioinformatics

**Genomic sequencing techniques.** Samples of whole bacterial isolates for whole genome sequencing were submitted to the Wellcome Sanger Institute, Hinxton. Half of the DNA from each sample remained in storage at −80 °C at MLW, and half were transferred via sterile pipette into a 0.3 ml sterile FluidX 2D Sequencing Tube (FluidX Ltd, UK).

Genomic sequencing libraries were prepared using the NEBNext Ultra II (New England Biolabs, Massachusetts, USA), multiplexed at 384 unique dual indexed barcode combinations, and sequenced on Illumina HiSeq X10 to generate 150 bp paired end reads. Post sequencing quality control showed a mean insert size of 180 bp and a mean fragment size of 450 bp. The median depth of coverage was 74.4. FastQC (version 0.11.9) and multiQC (version 0.11.8) were used to assess per base sequence quality, quality scores per sequence, per base sequence content, per base GC content, per sequence GC content, per base N content, contig length distribution and sequence duplication levels[67] (for cut-off values see Supplementary Table 4).

Read quality control was undertaken using Kraken (cut-off proportion reads <70% abundance *Salmonella*, Kraken version 1.1.1)[68]. CheckM (cut-offs used; contamination > 20% or completeness <90% removed, CheckM version 1.1.2) was run to assess contamination, strain heterogeneity and completeness of the genomes[69]. Assembly Statistics (genome length > 7 Mbp or contigs > 500 removed, Assembly Statistics version 1.0.1) was run to analyse the total genome length and number of contigs[70]. The Quality Assessment Tool for Genome Assemblies (QUAST)(cut-offs used contigs >500, N50 <20kbp or total base pairs <4Mbp or >5.8Mbp, QUAST version 5.0.2) was used to assess the number of contigs, N50 and total length of the genome[71] (Supplementary Table 4). Following the completion of quality control procedures, genomes were submitted to Pathogenwatch, which uses SISTR to assess the species, serovar and ST of the bacteria present[33,72]. One of the genomes which passed quality control procedures was incompletely assembled in the WSI pipeline (lane ID 34747_4#7), therefore SPADES (version 3.14) was used to assemble the genome and Prokka (version 1.14.5) was used for genome annotation[73,74]. In total 227 good quality whole genome sequences were identified which passed the stipulated thresholds.

**Core-genome phylogeny and SNP analysis.** A core and pangenome analysis was performed using Panaroo (version 1.3.3)[39]. A gene was considered core if it was present in 100% of the genomes at a match identity threshold of 98%[75]. A core genome sequence alignment was generated using Panaroo by concatenating the alignments of the core genes. Single nucleotide polymorphic (SNP) site alignment was

generated from the core genome alignment using SNP-sites (version 2.5.1)[76]. IQtree version 2.2.0) was run on the resulting core SNP-alignment to construct a maximum likelihood tree using the core gene SNP alignment of all 227 isolates[77]. Reliability of inferred branch partitions was assessed with 100 bootstrap replicates. The tree was visualised using ITOL (version 5) and ggtree (version 3.2)[78–80].

**Identification of AMR determinants, virulence factors and plasmid typing.** AMRFinderPlus (version 3.10) was used to detect chromosomal mutations encoding for AMR, acquired AMR genes (ARGs) and heavy metal resistance genes[81,82]. Those ARGs with an identity of 95% and a coverage of 95% were taken forward for further analysis.

**Determining appropriate thresholds for epidemiological analysis of putative bacterial sharing**

During the laboratory culture work, up to five picks of *Salmonella* were isolated from each positive sample to capture multi-serovar carriage and within-host diversity. To refine the collection to include solely genetically distinct *Salmonella* isolates from each host, pairwise SNP distance measurement using the core genome alignment was used to detect the SNP distance between any *Salmonella* isolates originating from the same individual.

Pairwise SNP distances were calculated using 'pairwise difference count' and snp-dists in order to measure the average number of SNP differences between strains within each sub-clade[83,84]. A SNP distance of 0 SNPs was used as a cut-off to define putative sharing of salmonellae, a 'sharing pair'[44].

**Epidemiological analysis of 'sharing-pairs'.** Systematic consideration of each sharing pair alongside the metadata was carried out and epidemiological links between sharing-pairs were established. Household-level sharing was defined as a sharing-pair of which both genomes within the pair originated from samples collected from different hosts within the same household. A between-household sharing-pair pairs was defined as a pair in which each of the isolates were collected from different households. IGraph (version 1.3.5) was used to visualise a network of the sharing pairs[85,86].

**Statistical methods**

Analysis was conducted using R version 2022.12.0 + 353[87]. Missing data were rare and unless otherwise specified missing variables were managed by exclusion from analysis.

**Ethics**

The study complies with all relevant ethical regulations. Ethical approval for this study was obtained from the University of Liverpool Veterinary Research Ethics Committee (Reference number VREC686) and the College of Medicine Research Ethics Committee (COMREC), Malawi (Reference Number P.02/18/2368). Informed, written consent was obtained from all household heads and individual household members (or their representatives) prior to their entry into the study following discussion of the study protocol, risks and benefits, financial and confidentiality considerations and details of methods to obtain more information. Should the prospective study participant be illiterate, the study was explained verbally and the consent form was read to the participant by the study team, witnessed by an additional neighbour who was not a member of the household. If the participant agreed to enter the study, the witness signed and dated the form and the witness documented their consent with an inked thumbprint. Parents or guardians were invited to consent for their children/wards of less than 18 years to join the study. Written assentwas sought for children between the ages of 8–18 in accordance with WHO guidelines[60]. For children younger than 8 years, the parent or guardian of the child was asked to provide full written consent. This research does not result in stigmatistion, incrimination, discrimination or otherwise personal risk to the participants. All human data has been anonymised. Benefit sharing measures have been discussed and agreed with the local host institution in Malawi, and at least one aliquot of all sample materials, including extracted DNA, remain in country.

**Reporting summary**

Further information on research design is available in the Nature Portfolio Reporting Summary linked to this article.

## Data availability

The data supporting the findings of this study are available in this article, the Source data file (Source Data- Circulation of *Salmonella* spp. between humans, animals and the environment in animal-owning households in Malawi) and have been deposited in the Zenodo data base under at the following https://doi.org/10.5281/zenodo.17191987. Raw sequencing reads for all novel sequences are deposited at the European Nucleotide Archive (ENA) under project (PRJEB32657). All accession numbers (both novel and previously published) used in this project are listed in the Source Data file ('Source Data- Circulation of *Salmonella* spp. between humans, animals and the environment in animal-owning households in Malawi') along with all metadata used for analysis in Figs. 1, 2, 3 and Supplementary Figs. 3, 4, 5 and 6. Previously published contextual metadata used in Supplementary Fig. 5 and Supplementary Fig. 6 are displayed in Source Data file ('Source Data– Circulation of *Salmonella* spp. between humans, animals and the environment in animal-owning households in Malawi'). Publically available sequence data was downloaded from one of the following sources: GenBank (https://www.ncbi.nlm.nih.gov/genbank/), Sequence Read Archive (https://www.ncbi.nlm.nih.gov/sra), European Nucleotide Archive (https://www.ebi.ac.uk/ena) or Enterobase (https://enterobase.warwick.ac.uk). Source data are provided with this paper.

## Code availability

The R code for the current study is publically available on GitHub at the following https://doi.org/10.5281/zenodo.1719187[88].

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

## Acknowledgements

We would like to thank the households recruited to this study for their participation in the sample collection. This work has been supported by a Wellcome Trust Clinical Fellowship award to Catherine N. Wilson, grant number 203919/Z/16/Z. M.A.B. and N.R.T. were supported by Wellcome Funding to the Sanger Institute (#206194). N.R.T., M.A.B., P.M. and C.N.W. were also supported by Wellcome Funding to the Sanger Institute (220540/Z/20/A) during the course of this work.

## Author contributions

C.N.W., E.M.F., N.A.F. and G.P. were involved in the conceptualisation and designing the overall study. C.N.W., J.N., A.M., M.D., L.B. and P.D. performed the fieldwork and sample collection. N.E. and C.J. assisted with study and laboratory protocol design. C.N.W. and L.M. designed the ODK data capture database and questionnaire and L.M. programmed this. C.N.W., Y.D., O.K., Z.K. and C.S. performed the microbiological and molecular laboratory work for this study. C.N.W. performed the bioinformatic analysis with the assistance of P.M., M.A.B., under the overall supervision of N.R.T. E.M.F., N.A.F. and G.P. supervised the entire study. C.N.W. wrote the paper and all of the authors commented on the paper draft. E.M.F. and N.R.T. are the co-senior authors of this paper.

## Competing interests

The authors declare no competing interests.
