## [Transparent Peer Review file · Nature Communications]

Circulation of *Salmonella* spp. between humans, animals and the environment in animal-owning households in Malawi

Corresponding Author: Dr Catherine Wilson

Version 0:

Reviewer comments:

Reviewer #1

(Remarks to the Author)

The study provided genomic epidemiology evidence for within and between households transmission of *Salmonella* among humans, domestic animals, and the environment in a less studied LMICs setting. The findings are expected given the practice of shared habitation of humans and animals in the setting. It was acknowledged that the reservoir and transmission route of iNTS pathogens, a much coveted question in the field, remained unanswered, perhaps due to the small sample size of the study. The acknowledgement is brief and without any discussion of previous studies that suggested foodborne transmission (<https://academic.oup.com/cid/article/73/7/e1570/5890413>) and human-to-human transmission (and no animal and environmental transmission) (<https://journals.plos.org/plosntds/article?id=10.1371/journal.pntd.0010982>) of iNTS.

L165: What does "gene flow within the collection" mean? It seems to imply HGT between strains within the collection, but no analysis was done to support the notion.

L229: Additional intraserotype comparisons of median pairwise SNP distances (within each major serotype found in all households) may provide higher resolution in characterizing transmission or persistence at the household scale than interserotype comparisons.

L305: This is more conservative than commonly used cluster definition for outbreak investigations. Also try a more relaxed transmission cutoff such as 5 SNPs?

Figure 3 has low resolution. It's barely legible. Was any isolate from wild birds involved in the transmission? Make it explicit in the text?

Reviewer #2

(Remarks to the Author)

The manuscript "Circulation of *Salmonella* spp. between humans, animals and the environment in urban and rural, low-intensity, animal-owning agricultural systems in Malawi" by Catherine N. Wilson et al details a longitudinal prospective study in which extensive stool and environmental sampling is carried out in two regions in Malawi—Ndirande (urban) and Chikwawa (rural)—to characterize transmission of *Salmonella* between humans, animals that live in and/or around the household, and the environment. This study identifies transmission pathways between a wide variety of carriers, both within and between households, with the rural site showing an increase in *Salmonella* presence and transmission occurrences. Overall, this is a good study which explores several interesting concepts (host diversity, low-intensity agriculture, urban vs. rural transmission), but there are some issues that need to be addressed.

Overall comments:

1. Page 1 – "strategy" misspelled in abstract
2. Page 1 – References 1 and 2 are 15 and 18 years old; suggest including more recent references to support these statements.
3. Page 2 – Would suggest adding to the sentence containing Reference 12: *Frontiers | Presence and Persistence of Salmonella in Water: The Impact on Microbial Quality of Water and Food Safety* ([frontiersin.org](https://www.frontiersin.org))
4. Page 2 – Reference 13 is the source from which the numbers in the preceding sentence are taken; however, Reference

- 13 is a 2010 study looking at 2004-2005 numbers. Authors should use a more recent source, or if one is not available, state the year in the text.
5. Page 2 –Reference 18 is 15 years old and seems to be specific to data from a single children’s hospital in Mozambique and finds that 26% (not 29%) of bacteremia cases in children at this hospital were NTS. This reference is not adequate to support the statement that “invasive NTS accounts for 29% of cases of bloodstream infections in Africa.” A better reference is needed or the text should be adjusted to reflect the study more accurately.
6. Page 2 – In addition to References 19 and 20, which are from 2009 and 2016, there are many more recent articles on the subject of ST313 and ST11 which should be included. See for example: The genomic epidemiology of multi-drug resistant invasive non-typhoidal Salmonella in selected sub-Saharan African countries - PMC (nih.gov)
7. Supplemental Figure 1 - Please label the triangles indicating Ndirande and Chikwawa in the map on the left. Also for this map, please indicate where the zoomed in portion is on the larger map.
8. Page 3 and 10, supplemental figure 2 – It is challenging to understand where multiple colony picks are used in this study, is it after the first round of selective broth, in Supp Fig 2 it mentions only 1 colony is used, then there are multiple colony picks according to the text after XLD round two before tr PCR, it is unclear why multiple colony picks are used if it is a single colony per sample frozen initially. Then multiple colony picks are sequenced as mentioned in the methods. Please clarify the methods regarding multiple colony picks and of the genomes sequenced is it one genome per sample or one genome per colony pick and ultimately multiple genomes per sample? If multiple genomes per sample, do you see different strains per sample?
9. Page 3 (and throughout paper) – Authors state there are 227 genomes used in the study, with 32 of these from Ndirande and 195 from Chikwawa. However, the dataset supplied in Supp. Data 1 indicates 33 Ndirande and 194 Chikwawa samples. Assuming the table is the correct version, the mislabeled genome would need to be identified, any pertinent analysis redone, and all figures/text updated to reflect the change.
10. Page 3 – The change from “87/965 (9.0%) samples from Ndirande and 146/1,115 (13.1%) samples from Chikwawa” at the bottom of the 2nd paragraph to “32/227 (14.1%) genomes originated from samples collected in Ndirande, while 195/227 (85.9%) originated from samples collected in the Chikwawa” is at first confusing to the reader, as the number of genomes for Chikwawa doubles from the number of samples (87 samples becomes 195 genomes). The authors later clarify that 2-3 colonies from each sample were sequenced, which is a sound methodology to discover in-host diversity and promote collection of at least one high quality genome. However, would recommend that this methodology be stated clearly within this section to prevent confusion and the number of replicates taken at each step be added to Supp. Figure 2.
- a. Authors should also clarify why there is such a drop for the Ndirande samples. Presumably the initial 87 samples had 2-3 colonies sequenced, allowing for the possibility for up to 261 genomes. However, the authors state that only 32 genomes were used from Ndirande (again, there are 33 in Supp. Data 1). Please explain why so many genomes were unusable for this study site.
- b. Additionally, authors need to clarify whether duplicates (cloned colonies from the same sample that were not unique in some way) are present in the 227 genomes. It appears from Suppl. Table 1 that there are duplicates among these genomes—for example sample numbers ERS4364137 and ERS4364138 appear to be identical. This would also mean that Figure 2’s phylogeny and household circles contain duplicates. If authors agree that these are duplicates, please explain why unique genomes were not preferable for these portions of the analysis.
11. Page 6, 2nd paragraph, last sentence – There are 13 animal species listed in the parentheses.
12. Page 6, 4th paragraph – Authors state “Given the previously documented increase in virulence genes and antimicrobial resistance and plasmids in Salmonella...” but Reference 28 does not seem to mention increases in virulence or plasmids.
13. Page 7 (bottom) to Page 8 (top) - Are "sample pairs" here the same as "genome pairs", which is the language used later and in Table 1? If they are the same, recommend sticking to one label for ease of reading--especially given the difference in meaning between samples and genomes introduced earlier in the paper.
14. Page 8, 2nd paragraph – Errant comma at the end of the first sentence.
15. Page 9, Figure 2:
- a. There are several inconsistencies between this graphic and the data provided in Supp. Data 1. As previously mentioned, Supp. Data 1 indicates 194 Chikwawa and 33 Ndirande isolates for a total of 227. However, Figure 2 has 208 filled circles for Chikwawa and 34 for Ndirande. Looking closer: CHH6 has 35 circles (36 in Supp. Data 1); CHH12 has 28 circles (10 in Supp. Data 1); CHH14 is not listed but there are 5 isolates in Supp. Data 1; CHH15 has 2 circles (15 in Supp. Data 1); NHH2 has 3 circles (2 in Supp. Data 1).
- b. Some of the green and pink lines indicating samples in the same household are wider than others. Please confirm if color choices are color blind friendly.
- c. CHH11 is missing a line.
- d. Figure legend indicates grey shading for isolates having no AMR markers, but it appears white. Cannot read scale bar, please add bootstrap values.
16. Page 11, Table 1 – Please clarify if the row entitled “Very closely related (10SNPs)” is actually less than or equal to or exactly 10 SNPs.
17. Page 14, Figure 3b – Cannot read due to low resolution. Would recommend adding the visit number to this graphic if possible and to include a graphical legend for the circle colors next to Figure 3b for ease of interpretation.
18. Page 15, last paragraph - The Ludden paper (Reference 33) does not seem to support this statement—it looks at E. coli nosocomial transmissions and does not seem to consider animals or the environment.
19. Page 16, last paragraph – “Africa” misspelled in second to last sentence.
20. Page 16, last paragraph, last sentence – There is at least one paper that explores the question posed by the authors (Case-control investigation of invasive Salmonella disease in Malawi reveals no evidence of environmental or animal transmission of invasive strains, and supports human to human transmission - PubMed (nih.gov)). Authors should consider conducting a literature review and considering how their work ties in with previous conclusions.
21. Page 17, Conclusions – Authors state “We show that sharing of identical salmonellae between humans, animals and the environment is common.” However, the present study only identified 20 genome pairs out of 227 genomes, which means

only 17.6% of the genomes were shared, which isn't really "common." Would recommend rephrasing to better reflect the findings.

22. Page 23, Controls – ">80\$" is probably meant to be ">80%".

23. Supplemental Figure 4:

a. Both figures are labeled 4b.

b. The legend colors do not match those in the charts.

c. Axis values are too small to read. Not sure what is on each axis and therefore difficult to read the figure.

Reviewer #3

(Remarks to the Author)

Reviewer #4

(Remarks to the Author)

Key results

This study examines the distribution of *Salmonella* in humans, animals and the environment in a small cohort of households in Malawi. The study found *Salmonella* in 11.2% of samples collected. Of the 227 salmonellae isolated, 125 were *S. enterica enterica* and 102 were *S. enterica salmae*. *Salmonella* were predominantly isolated from animal stool (75%). The study allowed an interesting examination of sharing salmonellae between animals and humans at a household level using whole genome sequencing.

Validity

The study appears valid in terms of methods and approach. I am concerned about the small number of good quality genomes from Ndirande, as it makes the representativeness questionable. What happened in this instance? Overall, the manuscript is interesting and well presented. I think there is an over-reliance on the term transmission, when what has been studied is carriage of salmonellae.

Significance

This is an excellent study that examines occurrence within the urban and rural setting in Africa. I would like to see more of these types of studies in other countries.

Data and methodology

1. Line 52. In the introduction, the authors mention that ~89% of salmonella are estimated to be foodborne, which seems high to me. Note that a global expert elicitation estimated that 46% of salmonella in Africa was related to foodborne transmission. See:

a. Hald T, Aspinall W, Devleeschauwer B, Cooke R, Corrigan T, Havelaar AH, Gibb HJ, Torgerson PR, Kirk MD, Angulo FJ, Lake RJ, Speybroeck N, Hoffmann S. World Health Organization Estimates of the Relative Contributions of Food to the Burden of Disease Due to Selected Foodborne Hazards: A Structured Expert Elicitation. *PLoS One*. 2016 Jan 19;11(1):e0145839. doi: 10.1371/journal.pone.0145839. PMID: 26784029; PMCID: PMC4718673.

2. Line 94. What was the success rate of collecting samples from humans, as it looks like less than one per person per visit. Please present data on isolation rates by age, as we would expect the majority of salmonellae in young children.

3. Line 95. In terms of sampling from households, what were the reasons for such high variability in numbers of samples collected?

4. I can't understand the distribution of genomes between the two study sites. In lines 104-5, 87 samples from Ndirande and 146 from Chikwawa are positive. In lines 112-3, it states that there 32 genomes from Ndirande and 195 from Chikwawa. How is there more genomes from Chikwawa than positive samples? Were there more than one colony picks off positive plates in some samples? If the different distribution relates to quality of sequencing, why was there such low quality in Ndirande?

5. Line 132, this multi-panel figure is a bit confusing as readers may think that the legend for part a relates to the remaining panels. I suggest separating part a and b. I had a hard time distinguishing the 64 different sequence types in the part b of the figure.

6. In line 151, the authors describe using Chi Square tests for testing differences between categories. For all statistical tests, it is important to consider using Fischer Exact if some of the expected values in cells are less than five.

7. Figure 3. Suggest using something other than a tree to denote environment, although I know this is tricky.

8. Line 416. I don't think the term 'the Global North' is helpful to distinguish groups of countries. Why not talk about 'industrialized' settings or 'high-income' countries, as you are really trying to say that proximity to animals leads to shared salmonellae. This definitely happens in high-income countries where close proximity occurs.

9. Line 421. I really don't like claims of primacy for things like this, as it doesn't really make it more important and it may not be true. There has been a lot of work done on sharing strains of salmonellae in the urban environment. In Australia, investigators found a high prevalence of shared salmonellae in the Darwin urban environment, which is tropical in nature. See:

a. Williams S, Markey P, Harlock M, Binns P, Gaggin J, Patel M. Individual and household-level risk factors for sporadic salmonellosis in children. *J Infect*. 2016 Jan;72(1):36-44. doi: 10.1016/j.jinf.2015.09.014. Epub 2015 Sep 28. PMID: 26416475.

b. Williams S, Patel M, Markey P, Muller R, Benedict S, Ross I, Heuzenroeder M, Davos D, Cameron S, Krause V. *Salmonella* in the tropical household environment—Everyday, everywhere. *J Infect*. 2015 Dec;71(6):642-8. doi: 10.1016/j.jinf.2015.09.011. Epub 2015 Sep 28. PMID: 26416474.

10. You don't really talk much about the most likely animal reservoirs of *S. salmae* Do you think there were unsampled

animal reservoirs, such as snakes, frogs and other lizards in the study? Is this worth further investigation.

11. Line 458. Do you know if *S. salmae* causes much human disease in terms of gastroenteritis and invasive infections in Malawi?

12. Line 464. Throughout the manuscript the authors talk about 'transmission'. Here it mentions 'transmission between households'. In reality, this study is not documenting transmission, but point prevalence. I feel that the use of transmission is too strong, as it implies that one gave it to the other, which isn't really a correct interpretation. The same strain was isolated from multiple households or species.

13. Line 475. I think that the sharing of strains between dogs and poultry isn't that interesting, it is expected.

14. Line 483. The authors mention that animal faeces removal and hand hygiene is 'profoundly important' in reducing spread of salmonellae. I think it is important not to overplay the fact that shared strains in different hosts were identified, but the connection to disease and transmission is less clear. This also begs the question about cleanliness of food preparation areas and food storage, which are important.

15. Supplementary information: I note a high prevalence of *Salmonella* Gaminara. Is this serotype known to cause disease in humans? Are any of the other serotypes less pathogenic to humans? It would be good to display this information by host species. I also note a high diversity of *S. salmae* serotypes as well.

Version 1:

Reviewer comments:

Reviewer #4

(Remarks to the Author)

I appreciate the response of the authors to my queries. I have no additional questions or queries.

(Remarks on code availability)

REVIEWER COMMENTS

Reviewer #1 (Remarks to the Author):

The study provided genomic epidemiology evidence for within and between households transmission of *Salmonella* among humans, domestic animals, and the environment in a less studied LMICs setting. The findings are expected given the practice of shared habitation of humans and animals in the setting. It was acknowledged that the reservoir and transmission route of iNTS pathogens, a much coveted question in the field, remained unanswered, perhaps due to the small sample size of the study. The acknowledgement is brief and without any discussion of previous studies that suggested foodborne transmission (<https://academic.oup.com/cid/article/73/7/e1570/5890413>) and human-to-human transmission (and no animal and environmental transmission) (<https://journals.plos.org/plosntds/article?id=10.1371/journal.pntd.0010982>) of iNTS.

Authors' Response: We agree that the question of the reservoir of iNTS pathogens remains unanswered within this study. The acknowledgement is brief, as although this question is important, this is not the main focus of this manuscript. We have cited the important papers which you have suggested here within the paper on 569 with the sentence;

Line 570: 'This is predominantly consistent with previous findings, however, the potential for foodborne transmission cannot be discounted, as *S. Enteritidis* ST11 of the global epidemic clade has been previously isolated from samples collected within the livestock and poultry meat pathway in Tanzania⁵¹⁻⁵³.

L165: What does "gene flow within the collection" mean? It seems to imply HGT between strains within the collection, but no analysis was done to support the notion.

Authors' Response: In this instance, 'gene flow within the collection' refers to virulence genes, antimicrobial resistance determinants and plasmids within the collection of *Salmonella*. This has been clarified within this text with alteration of this subheading to;

Line 205: 'Inferring the distribution of plasmids, antimicrobial resistance determinants and virulence genes between bacterial isolates sequenced in this collection'

L229: Additional intra-serotype comparisons of median pairwise SNP distances (within each major serotype found in all households) may provide higher resolution in characterizing transmission or persistence at the household scale than inter-serotype comparisons.

Authors' Response: We did trial using intra-serotype comparison rather than interserotype comparison. However, as illustrated within Figure 1, the diversity of *Salmonella* detected across households was immense. Had a 'major serotype' been present repeatedly at numerous households, intra-serotype comparison would indeed have been a method which we would have employed. Since there are very few cases in this study where that was true, we have made inter-serotype comparisons.

L305: This is more conservative than commonly used cluster definition for outbreak investigations. Also try a more relaxed transmission cutoff such as 5 SNPs?

Authors' Response: The use of SNP thresholds is typically species and dataset dependent, and there is currently no published consensus cutoff for the pairwise SNP differences between *Salmonella* isolate genomes which has been used to infer transmission between hosts. Indeed, this cutoff value was something that we spent a long time considering. Since we used a core gene alignment for this cross-species analysis – an approach expected to result in a loss of sensitivity due to exclusion of intergenic regions, which is not the case when you use a reference based lineage analysis, we selected the most conservative choice in order to provide confidence that the putative transmissions we detected reflected recent events.

Figure 3 has low resolution. It's barely legible. Was any isolate from wild birds involved in the transmission? Make it explicit in the text?

Authors' Response: Thank you for pointing this out. We have altered Figure 3 to a high-resolution pdf which is more legible.

In order to clarify that we have not detected any isolate from wild birds to be involved in transmission, we have added the following line to the discussion:

Line 437: 'There was no sharing detected between wild birds and other hosts within the study.'

Reviewer #2 (Remarks to the Author):

The manuscript "Circulation of *Salmonella* spp. between humans, animals and the environment in urban and rural, low-intensity, animal-owning agricultural systems in Malawi" by Catherine N. Wilson et al details a longitudinal prospective study in which extensive stool and environmental sampling is carried out in two regions in Malawi—Ndirande (urban) and Chikwawa (rural)—to characterize transmission of *Salmonella* between humans, animals that live in and/or around the household, and the environment. This study identifies transmission pathways between a wide variety of carriers, both within and between households, with the rural site showing an increase in *Salmonella* presence and transmission occurrences.

Overall, this is a good study which explores several interesting concepts (host diversity, low-intensity agriculture, urban vs. rural transmission), but there are some issues that need to be addressed.

We thank the reviewer for their positive remarks about our study and goals.

Overall comments:

1. Page 1 – "strategy" misspelled in abstract

Thank you, this spelling has been corrected on line 22.

2. Page 1 – References 1 and 2 are 15 and 18 years old; suggest including more recent references to support these statements.

Thank you for this comment, despite their age these references are still valid and remain definitive literature on this topic. To address this point and while retaining reference to this seminal work, we have added ref Shaheen et al. (2022) and Nandi et al. (2021) to include more recently published articles.

Line 28, references 3 and 4:

Shaheen, M. N. F. The concept of one health applied to the problem of zoonotic diseases. *Rev. Med. Virol.* **32**, e2326 (2022).

Nandi, A. & Allen, L. J. S. Probability of a zoonotic spillover with seasonal variation. *Infect. Dis. Model.* **6**, 514–531 (2021).

3. Page 2 – Would suggest adding to the sentence containing Reference 12: Frontiers | Presence and Persistence of Salmonella in Water: The Impact on Microbial Quality of Water and Food Safety (frontiersin.org)

Thank you, this change has been made.

4. Page 2 – Reference 13 is the source from which the numbers in the preceding sentence are taken; however, Reference 13 is a 2010 study looking at 2004-2005 numbers. Authors should use a more recent source, or if one is not available, state the year in the text.

Authors' Response: We have updated to reference the figures published within the 'World Health global estimates and regional comparisons of the burden of foodborne disease in 2010', which was published in 2015.

Line 50: "Worldwide, *Salmonella* spp. are estimated to cause 78.7 million human cases of gastroenteritis annually, with 59,100 deaths and 4.1 million disability adjusted life-years (DALYs)".

Havelaar, A. H. *et al.* World Health Organization Global Estimates and Regional Comparisons of the Burden of Foodborne Disease in 2010. *PLOS Med.* **12**, e1001923 (2015).

5. Page 2 –Reference 18 is 15 years old and seems to be specific to data from a single children's hospital in Mozambique and finds that 26% (not 29%) of bacteraemia cases in children at this hospital were NTS. This reference is not adequate to support the statement that "invasive NTS accounts for 29% of cases of bloodstream infections in Africa." A better reference is needed or the text should be adjusted to reflect the study more accurately.

Thank you for this comment. These figures have been updated to the most recent literature. As such, the following, more up to date, references have been added:

Line 58, references 21 and 22:

Marchello, C. S., Dale, A. P., Pisharody, S., Rubach, M. P. & Crump, J. A. A Systematic Review and Meta-analysis of the Prevalence of Community-Onset Bloodstream Infections among Hospitalized Patients in Africa and Asia. *Antimicrob. Agents Chemother.* **64**, 10.1128/aac.01974-19 (2019).

Balasubramanian, R. *et al.* The global burden and epidemiology of invasive non-typhoidal *Salmonella* infections. *Hum. Vaccines Immunother.* **15**, 1421–1426 (2018).

6. Page 2 – In addition to References 19 and 20, which are from 2009 and 2016, there are

many more recent articles on the subject of ST313 and ST11 which should be included. See for example: **The genomic epidemiology of multi-drug resistant invasive non-typhoidal Salmonella in selected sub-Saharan African countries - PMC (nih.gov)**

Thank you for this comment. I have added in the following references to include some more recent work:

Line 60, references 23, 24, 27 and 28:

Park, S. E. *et al.* The genomic epidemiology of multi-drug resistant invasive non-typhoidal Salmonella in selected sub-Saharan African countries. *BMJ Glob. Health* **6**, e005659 (2021).

Akullian, A. *et al.* Multi-drug resistant non-typhoidal Salmonella associated with invasive disease in western Kenya. *PLoS Negl. Trop. Dis.* **12**, e0006156 (2018).

Pulford, C. V. *et al.* Stepwise evolution of Salmonella Typhimurium ST313 causing bloodstream infection in Africa. *Nat. Microbiol.* **6**, 327–338 (2021).

Van Puyvelde, S. *et al.* An African Salmonella Typhimurium ST313 sublineage with extensive drug-resistance and signatures of host adaptation. *Nat. Commun.* **10**, 4280 (2019).

7. Supplemental Figure 1 - Please label the triangles indicating Ndirande and Chikwawa in the map on the left. Also for this map, please indicate where the zoomed in portion is on the larger map.

Thank you for this comment, this change has been made.

8. Page 3 and 10, supplemental figure 2 – It is challenging to understand where multiple colony picks are used in this study, is it after the first round of selective broth, in Supp Fig 2 it mentions only 1 colony is used, then there are multiple colony picks according to the text after XLD round two before *ttr* PCR, it is unclear why multiple colony picks are used if it is a single colony per sample frozen initially. Then multiple colony picks are sequenced as mentioned in the methods. Please clarify the methods regarding multiple colony picks and of the genomes sequenced is it one genome per sample or one genome per colony pick and ultimately multiple genomes per sample? If multiple genomes per sample, do you see different strains per sample?

Authors' Response: We apologise for the confusion. It is not the case that a single colony per sample was taken from selective broth prior to taking multiple colony picks following the growth of a single colony. The method comprises of a loop of broth solution containing bacteria taken from Rappaport Vassiliadis enrichment (following buffered peptone water enrichment) and plated onto XLD and CASE agar. We have altered Suppl. Figure 2 to clarify the number of isolates/picks which were used at each stage of the processing.

This is an outline of the methods used:

Samples were taken from human, animals and environmental sources and following enrichment in buffered peptone water and Rappaport Vassiliadis broths a loop of bacteria were streaked onto each of XLD and CASE agar plates. From the XLD and CASE agar culture plates up to 5 colonies per sample were randomly selected and frozen in cryopreservative (individual microbanks or modified microbank tubes), of which two to three colonies were sequenced.

From sequencing multiple colonies for each sample there were examples of multiple strains being present in the same sample.

For example, the following genomes were collected from the same host, at the same visit and provided identical genomes (as confirmed by pairwise SNP analysis):

- 34124_7#12 and 34124_8#5
- 34124_8#58 and 34124_7#64
- 34124_8#39 and 34124_8#37

Additionally, we did see different strains present within the same sample, for example:

- 34124_8#203 (*S. Concord*) and 34124_8#239 (*S. Enteritidis*) were detected within the faeces of a single dog within household CHH6 at the same visit to the household.
- 34873_1#186 (*S. Poona*) and 34124_8#98 (*S. Johannesburg*) were detected within the faeces of a single chicken within household CHH10 at the same visit to the household.
- 34124_8#158 (*S. Gaminara*) and 34124_8#155 (*S. salamae* II 6,7:z:z6) were detected within the faeces of a single gecko within household CHH12 at the same visit to the household.

The methods have been clarified in the text on page 20 with insertions highlighted in bold:

Line 680: 'Salmonellae were isolated and identified by selective culture using enrichment steps using buffered peptone water and Rappaport Vassiliadis for 24 hours each respectively and **a loop of bacterial solution from each were streaked out** onto each of CASE and XLD selective agar plates prior to incubation at 37°C (Suppl. Figure 2). **From the XLD and CASE agar culture plates up to 5 colonies per sample were randomly selected and underwent** O and Vi antigen testing to confirm the presence of salmonellae, and the absence of typhoidal-*Salmonella*. Once isolated, aliquots of pure bacteria growth were stored at -80°C in individual microbanks or modified microbank tubes. Up to five picks of suspected *Salmonella* isolates from each sample were stored.'

Line 697: 'DNA intended for whole genome sequencing was extracted from *Salmonella* isolates using commercial kits (Qiagen DNA Mini kit). DNA of **two-three frozen colonies per sample** was quantified using a Qubit© (Thermo Fisher Scientific, MA, USA) and then sent for whole genome sequencing at the Wellcome Sanger Institute, UK'.

9. Page 3 (and throughout paper) – Authors state there are 227 genomes used in the study, with 32 of these from Ndirande and 195 from Chikwawa. However, the dataset supplied in Supp. Data 1 indicates 33 Ndirande and 194 Chikwawa samples. Assuming the table is the correct version, the mislabelled genome would need to be identified, any pertinent analysis redone, and all figures/text updated to reflect the change.

Thank you for noticing this error. Suppl. Data 1 is correct; there are 33 Ndirande and 194 Chikwawa samples. Within this dataset genome with lane id 404441_1#69 has been correctly labelled as from NHH10 in Ndirande, collected from a dog. There was an error on a duplicate spreadsheet which we used to calculate the figures written in the text on page 3. This error was NOT carried through to the remainder of the paper, as outlined below, and in the present version, the figures are all correct.

1. 404441_1#69 was collected within the same faecal sample from the same female dog, at the same visit, as 34124_7#185. Both of these genomes are *S. enterica* subsp. *enterica* spp. Oranienburg.
2. Figure 1 is correct (there are 2 genomes detected from samples collected from NHH10, both from a female dog).
3. Figure 2 is correct (both genomes have been included and correctly labelled as from Ndirande).
4. Referring to the pairwise SNP distance of the core gene comparison, these two genomes are 0 SNP distance apart and as they originate from the same host and were collected at the same visit (within the same faecal sample) one genome from the pair has been removed for the downstream sharing pairs analysis.

One metadata error which has been carried through is the AMR determinant profile of the two genomes depicted in Figure 2. Both 34124_7#185 and 404441_1#69 carry *fosA7* within their genomes; Figure 2 has been replaced to correct this. The study site location of the two genomes is correct in Figure 2.

10. Page 3 – The change from “87/965 (9.0%) samples from Ndirande and 146/1,115 (13.1%) samples from Chikwawa” at the bottom of the 2nd paragraph to “32/227 (14.1%) genomes originated from samples collected in Ndirande, while 195/227 (85.9%) originated from samples collected in the Chikwawa” is at first confusing to the reader, as the number of genomes for Chikwawa doubles from the number of samples (87 samples becomes 195 genomes). The authors later clarify that 2-3 colonies from each sample were sequenced, which is a sound methodology to discover in-host diversity and promote collection of at least one high quality genome. However, would recommend that this methodology be stated clearly within this section to prevent confusion and the number of replicates taken at each step be added to Supp. Figure 2.

Thank you for considering this, and for the comments that are supportive of the way in which we chose to select isolates for sequencing. We have added in the following sentence to the first line of this paragraph which aims to clarify the sequencing strategy.

Line 122:

“Whole genome sequencing was used to confirm the presence of *Salmonella* spp. We performed DNA extraction and whole genome sequencing of two-three *ttr* positive colony picks per sample to enable us to assess multi-serovar carriage and within-host diversity. We therefore sequenced 403 isolates in total, which were taken from a total of 214 samples. After quality assessment of sequence data, 227 genomes were identified as *Salmonella* genomes, passing quality thresholds, and were subsequently included in our analysis (Figure 1A). These 227 genomes originated from 111 discrete samples. More than one individual *Salmonella* isolate was identified from twenty of these samples (within-host diversity).”

We have also added the number of replicates taken at each step to Supp. Figure 2.

a. Authors should also clarify why there is such a drop for the Ndirande samples. Presumably the initial 87 samples had 2-3 colonies sequenced, allowing for the possibility for up to 261 genomes. However, the authors state that only 32 genomes were used from Ndirande (again,

there are 33 in Supp. Data 1). Please explain why so many genomes were unusable for this study site.

Thank you for this comment. Yes, your calculations are correct. We have corrected the mistake within the text; the Supplementary Data is correct there were 33 genomes from Ndirande.

Within the text we have clarified the number of samples and number of individual genomes found to be containing *Salmonella* following whole genome sequencing in both Ndirande and Chikwawa. The PCR test for the *ttr* gene, which was used as a screening test for *Salmonella*, suggested that 233/2,080 samples contained *Salmonella* (87/965 (9.0%) from Ndirande and 146/1,115 (13.1%) from Chikwawa). In total 403 isolates originating from these 233 samples were submitted for whole genome sequencing.

Of these isolates, whole genome sequencing confirmed the presence of *Salmonella* in 227 isolates which originated from 111 samples; 94 samples collected from Chikwawa and 17 samples collected from Ndirande. Therefore, in total 94/1,115 (8.4%) of samples collected from Chikwawa and 17/965 (1.8%) of samples collected in Ndirande contained *Salmonella*.

The vast majority of the isolates which were not *Salmonella* positive following whole genome sequencing were alternative bacteria, particularly *Proteus* spp., rather than poor quality *Salmonella* genomes. Whole genome sequencing has demonstrated that the prevalence of *Salmonella* was lower in the urban area Ndirande than in the rural area, Chikwawa, nevertheless the samples remain representative of the study population.

We have explained this in the following paragraphs which have been modified on pages 3 and 4 of the manuscript:

Line 114-155:

'In total, 2,080 individual samples were collected, 965 (46.4%) from Ndirande and 1,115 (53.6%) from Chikwawa. Samples were cultured and the presence of *Salmonella* was screened for by PCR for *ttr*. PCR for *ttr* was positive in 233/2,080 samples (11.2%) (Suppl. Figure 2, Suppl. Table 1-3). In total 87/965 (9.0%) of samples from Ndirande and 146/1,115 (13.1%) of samples from Chikwawa were positive for *Salmonella* spp..

Distribution of *Salmonella* genomes

Whole genome sequencing was used to confirm the presence of *Salmonella* spp. We performed DNA extraction and whole genome sequencing of two-three *ttr* positive colony picks per sample to enable us to assess multi-serovar carriage and within-host diversity. We therefore sequenced 403 isolates in total, which were taken from a total of 214 samples. After quality assessment of sequence data, 227 genomes were identified as *Salmonella* genomes, passing quality thresholds, and were subsequently included in our analysis (Figure 1A). These 227 genomes originated from 111 discrete samples.

On examination, this collection of 227 genomes contained a number of 'identical' bacterial isolates (i.e. identical bacterial isolates are detected within one sample, collected at the same household at the same timepoint). One isolate from a pair of identical isolates was removed (de-duplicated) from this collection, leaving a total of 131 individual isolates (an individual isolate is a single

Salmonella isolate of each serovar or sequence type from each individual sample). These 131 individual isolates were collected from 111 samples, therefore more than one individual *Salmonella* isolate was identified from twenty of these samples (within-host diversity). The 111 samples originate from 81 animal stool samples (73.0%), 16 environment samples (14.4%) and 14 human stool samples (12.6%). At least one *Salmonella* genome was generated from 25/30 households in the study; 14/15 (93.3%) households sampled in Chikwawa and 11/15 (73.3%) households sampled in Ndirande (Figure 1A).

In total 94/1,115 (8.4%) of samples collected from Chikwawa and 17/965 (1.8%) of samples collected in Ndirande contained *Salmonella*.

b. Additionally, authors need to clarify whether duplicates (cloned colonies from the same sample that were not unique in some way) are present in the 227 genomes. It appears from Suppl. Table 1 that there are duplicates among these genomes—for example sample numbers ERS4364137 and ERS4364138 appear to be identical. This would also mean that Figure 2’s phylogeny and household circles contain duplicates. If authors agree that these are duplicates, please explain why unique genomes were not preferable for these portions of the analysis.

Thank you for this comment. This is correct, duplicates have been included within the collection described within Suppl. Table 1 and Figure 2.

This study had multiple goals, which included describing the genomic diversity in the collection, as well as describing the genomic diversity within households and within individuals. As the reviewer has already noted, we selected and sequenced multiple single colony picks from individuals, as this allowed us to describe the genomic diversity within these individuals. Where genomes from the same individual belonged to different species or serotypes, including both in the analyses clearly addresses to diversity within both the individual and within the broader collection. Similarly, where multiple colony picks were nearly identical within an individual, this is also informative about diversity within an individual (whilst arguably being less informative about diversity in the collection).

Due to this added value of being able to describe patterns of both diversity and clonality in households and individuals, we intentionally retained duplicate colonies in our initial analysis. Figure 2 clearly shows instances where individuals samples from a household are highly similar, with other samples from the same household being phylogenetically distinct. This then led to the subsequent analyses including inferring transmission between hosts/niches, and here we deduplicated identical genomes to remove potential biases from this intentional oversampling.

11. Page 6, 2nd paragraph, last sentence – There are 13 animal species listed in the parentheses.

Thank you for pointing this out. ‘sheep’ was included in error this has been corrected.

12. Page 6, 4th paragraph – Authors state “Given the previously documented increase in virulence genes and antimicrobial resistance and plasmids in *Salmonella*...” but Reference 28 does not seem to mention increases in virulence or plasmids.

Thank you. To clarify this point we have changed this sentence on line 208:

‘Given the previously documented increase in antimicrobial resistance in *Salmonella* in this setting(1), we investigated gene flow of antimicrobial resistance determinants, as well as plasmids and virulence genes.’

13. Page 7 (bottom) to Page 8 (top) - Are "sample pairs" here the same as "genome pairs", which is the language used later and in Table 1? If they are the same, recommend sticking to one label for ease of reading--especially given the difference in meaning between samples and genomes introduced earlier in the paper.

Thank you very much for pointing this out. This was a mistake it should have read ‘pairs of genomes’ rather than ‘samples’. We have altered the text on line 284 to amend this.

14. Page 8, 2nd paragraph – Errant comma at the end of the first sentence.

Thank you, this has been removed.

15. Page 9, Figure 2:

a. There are several inconsistencies between this graphic and the data provided in Supp. Data 1. As previously mentioned, Supp. Data 1 indicates 194 Chikwawa and 33 Ndirande isolates for a total of 227. However, Figure 2 has 208 filled circles for Chikwawa and 34 for Ndirande. Looking closer: CHH6 has 35 circles (36 in Supp. Data 1); CHH12 has 28 circles (10 in Supp. Data 1); CHH14 is not listed but there are 5 isolates in Supp. Data 1; CHH15 has 2 circles (15 in Supp. Data 1); NHH2 has 3 circles (2 in Supp. Data 1).

Thank you very much indeed for taking the time to look at this in depth. This has now been corrected.

b. Some of the green and pink lines indicating samples in the same household are wider than others. Please confirm if color choices are color blind friendly.

Thank you, this has been corrected. The colour choices are now colour blind friendly.

c. CHH11 is missing a line.

Thank you, this has been corrected.

d. Figure legend indicates grey shading for isolates having no AMR markers, but it appears white. Cannot read scale bar, please add bootstrap values.

Thank you, this has been corrected.

16. Page 11, Table 1 – Please clarify if the row entitled “Very closely related (10SNPs)” is actually less than or equal to or exactly 10 SNPs.

Thank you, this has been done.

17. Page 14, Figure 3b – Cannot read due to low resolution. Would recommend adding the visit number to this graphic if possible and to include a graphical legend for the circle colors next to Figure 3b for ease of interpretation.

Thank you for this comment. Figure 3 has been changed to a pdf file which has increased the resolution. I have also added in a graphical key for Figure 3b and added the visit number to the graphic.

18. Page 15, last paragraph - The Ludden paper (Reference 33) does not seem to support this statement—it looks at *E. coli* nosocomial transmissions and does not seem to consider animals or the environment.

Thank you. Unfortunately, we incorrectly listed a different paper by Catherine Ludden and colleagues from the one we intended to,

Reference 42, Line 505: (Ludden, C. *et al.* One Health Genomic Surveillance of *Escherichia coli* Demonstrates Distinct Lineages and Mobile Genetic Elements in Isolates from Humans versus Livestock. *mBio* **10**, 10.1128/mbio.02693-18 (2019))

and have now corrected this, as well as adding a citation to the following recent studies by Day et al, Thorpe et al and Gouliouris *et al.* (Line 505, reference 40, 41 and 43):

Thorpe, H. A. *et al.* A large-scale genomic snapshot of *Klebsiella* spp. isolates in Northern Italy reveals limited transmission between clinical and non-clinical settings. *Nat. Microbiol.* **7**, 2054–2067 (2022).

Gouliouris, T. *et al.* Genomic Surveillance of *Enterococcus faecium* Reveals Limited Sharing of Strains and Resistance Genes between Livestock and Humans in the United Kingdom. *mBio* **9**, e01780-18 (2018).

Day, M. J. *et al.* Extended-spectrum β -lactamase-producing *Escherichia coli* in human-derived and foodchain-derived samples from England, Wales, and Scotland: an epidemiological surveillance and typing study. *Lancet Infect. Dis.* **19**, 1325–1335 (2019).

19. Page 16, last paragraph – “Africa” misspelled in second to last sentence.

Thank you, this has been altered.

20. Page 16, last paragraph, last sentence – There is at least one paper that explores the question posed by the authors (Case-control investigation of invasive *Salmonella* disease in Malawi reveals no evidence of environmental or animal transmission of invasive strains, and supports human to human transmission - PubMed (nih.gov)). Authors should consider conducting a literature review and considering how their work ties in with previous conclusions.

Thank you for your comment. Following consideration, the last sentence in the last paragraph on page 16 has been removed.

The sentence is referred to transmission of *Salmonella* as a genus as a whole, rather than just invasive strains of *Salmonella*. In this paper we discuss the broader ecology of the genus of *Salmonella*, rather than solely *Salmonella enterica* subsp. *enterica* serotype Typhimurium ST313. It is likely that the dynamics of *Salmonella* are different for different serovars, as indicated in our study.

Several other excellent studies have interrogated the transmission of *Salmonella enterica* subsp. *enterica* serotype Typhimurium ST313 in various settings in sub-Saharan Africa, and have not found strong evidence for animal or environment to human transmission(2,3). One study in the Democratic Republic of Congo has detected *S. Typhimurium* ST313 carried by rats within slaughterhouses in Kisangani, and identified that the ST313 isolates carried by rats were of Lineage 2 and multidrug resistant (5). The work presented in our study uses the resolution offered by whole genome sequencing to demonstrate that the ST313 detected within a dog was of a different lineage to that causing invasive disease in humans. Therefore, the question of the nature of the reservoir and key transmission routes of *S. Typhimurium* ST313 is still open.

Available evidence suggests direct human-human as the most likely transmission route of this invasive *S. Typhimurium* sequence type. Larger, detailed studies are necessary to clarify this point, however they are extremely challenging and expensive

The paragraph beginning line 559 details our specific findings in regard to *S. Typhimurium* ST313 in this study. I have lengthened the following sentence within this paragraph to clarify how our findings relating to the transmission of *S. Typhimurium* ST313 in this study correlate with work previously published by Koolman (2022) and Post (2019):

‘Within this study we have detected strong epidemiological links between *Salmonella* of relevance to human disease, but not to pathovars strongly associated with iNTS disease in Africa. This is predominantly consistent with previous findings.....⁵¹⁻⁵³.’

21. Page 17, Conclusions – Authors state “We show that sharing of identical salmonellae between humans, animals and the environment is common.” However, the present study only identified 20 genome pairs out of 227 genomes, which means only 17.6% of the genomes were shared, which isn’t really “common.” Would recommend rephrasing to better reflect the findings.

Thank you for this comment. We have amended this sentence in the conclusion (line 618) and now reads;

“We show that sharing of identical salmonellae between humans, animals and the environment is possible and in fact likely, demonstrating the importance of considering all aspects of hygiene and biosecurity precautions within households when developing strategies to limit the movement, carriage and sharing of salmonellae and other gastrointestinal pathogens.

It may have been expected that a relatively small sample size could completely miss any genome sharing. We do acknowledge that given that the sample size is small, we cannot quantify this sharing and that investigation of a larger study population would be advised to investigate this further. We have reflected upon this with the following paragraph within the discussion:

Line 595: ‘The total number of genome pairs detected within this study is low and therefore it is not possible to further quantify the sharing which has occurred. However, given the small number of total genomes (n=227) amongst which sharing was investigated, it may have been expected that sharing could have been missed entirely. Therefore, investigation of a larger study population would be warranted, in order to further quantify the sharing of *Salmonella* within and between households.’

22. Page 23, Controls – “>80\$” is probably meant to be “>80%”.

Thank you, this has been corrected.

23. Supplemental Figure 4:

a. Both figures are labelled 4b.

Thank you, done.

b. The legend colors do not match those in the charts.

Thank you, done.

c. Axis values are too small to read. Not sure what is on each axis and therefore difficult to read the figure.

Thank you, done.

Reviewer #3 (Remarks to the Author):

Reviewer #4 (Remarks to the Author):

Key results

This study examines the distribution of *Salmonella* in humans, animals and the environment in a small cohort of households in Malawi. The study found *Salmonella* in 11.2% of samples collected. Of the 227 salmonellae isolated, 125 were *S. enterica* subsp. *enterica* and 102 were *S. enterica* subsp. *salamae*. *Salmonella* were predominantly isolated from animal stool (75%). The study allowed an interesting examination of sharing salmonellae between animals and humans at a household level using whole genome sequencing.

Validity

The study appears valid in terms of methods and approach. I am concerned about the small number of good quality genomes from Ndirande, as it makes the representativeness questionable.

What happened in this instance?

Thank you for this comment.

Our screening test using *ttr* suggested the presence of *Salmonella* in 233/2,080 samples, 87/965 (9.0%) of which were from Ndirande and 146/1,115 (13.1%) of which were from Chikwawa. Whole genome sequencing was carried out on two-three colony picks per sample, in most cases, to assess within host diversity; therefore in total DNA from 403 isolates was submitted for whole genome sequencing. Whole genome sequencing confirmed the presence of *Salmonella* in 227 isolates (194/227 (85.5%) were from Chikwawa and 33/227 (14.5%) from Ndirande). These isolates originated from 111 samples. Therefore, in total 94/1,115 (8.4%) of samples collected from Chikwawa and 17/965 (1.8%) of samples collected in Ndirande contained *Salmonella*.

The vast majority of these isolates which were not *Salmonella* positive were alternative bacteria, particularly *Proteus* spp., rather than poor quality *Salmonella* genomes. There was therefore more *Salmonella* present in Chikwawa than Ndirande. We found that the prevalence of *Salmonella* was lower in the urban area Ndirande than in the rural area, Chikwawa, nevertheless the samples remain representative of the study population.

The text has been altered to clarify this point, between lines 114-155:

‘In total, 2,080 individual samples were collected, 965 (46.4%) from Ndirande and 1,115 (53.6%) from Chikwawa. Samples were cultured and the presence of *Salmonella* was screened for by PCR for *ttr*. PCR for *ttr* was positive in 233/2,080 samples (11.2%) (Suppl. Figure 2, Suppl. Table 1-3). In

total 87/965 (9.0%) of samples from Ndirande and 146/1,115 (13.1%) of samples from Chikwawa were positive for *Salmonella* spp..

Distribution of *Salmonella* genomes

Whole genome sequencing was used to confirm the presence of *Salmonella* spp. We performed DNA extraction and whole genome sequencing of two-three *ttr* positive colony picks per sample to enable us to assess multi-serovar carriage and within-host diversity. We therefore sequenced 403 isolates in total, which were taken from a total of 214 samples. After quality assessment of sequence data, 227 genomes were identified as *Salmonella* genomes, passing quality thresholds, and were subsequently included in our analysis (Figure 1A). These 227 genomes originated from 111 discrete samples.

On examination, this collection of 227 genomes contained a number of 'identical' bacterial isolates (i.e. identical bacterial isolates are detected within one sample, collected at the same household at the same timepoint). Identical isolates were de-duplicated from this collection, leaving a total of 131 individual isolates (an individual isolate is a single *Salmonella* isolate of each serovar or sequence type from each individual sample). These 131 individual isolates were collected from 111 samples, therefore more than one individual *Salmonella* isolate was identified from twenty of these samples (within-host diversity). The 111 samples originate from 81 animal stool samples (73.0%), 16 environment samples (14.4%) and 14 human stool samples (12.6%). At least one *Salmonella* genome was generated from 25/30 households in the study; 14/15 (93.3%) households sampled in Chikwawa and 11/15 (73.3%) households sampled in Ndirande (Figure 1A).

In total 94/1,115 (8.4%) of samples collected from Chikwawa and 17/965 (1.8%) of samples collected in Ndirande contained *Salmonella*.

We have added a paragraph to the Discussion to describe the limitation of the small sample size:

Line 595: 'The total number of genome pairs detected within this study is low and therefore it is not possible to further quantify the sharing which has occurred. However, given the small number of total genomes (n=227) overall, and in Ndirande (n=33) specifically, amongst which sharing was investigated, it may have been expected that sharing could have been missed entirely. Therefore, investigation of a larger study population is warranted, in order to further quantify the sharing of *Salmonella* within and between households, study sites and hosts.'

Overall, the manuscript is interesting and well presented. I think there is an over-reliance on the term transmission, when what has been studied is carriage of salmonellae.

Thank you for this important point. We have changed the language throughout the manuscript to discuss putative sharing and flux of *Salmonella*, rather than direct transmission. The similarity of the genomes does raise the possibility for transmission, but we agree that we cannot document absolutely that this has occurred.

Significance

This is an excellent study that examines occurrence within the urban and rural setting in Africa. I

would like to see more of these types of studies in other countries.

Thank you for your kind words.

Data and methodology

1. Line 52. In the introduction, the authors mention that ~89% of salmonella are estimated to be foodborne, which seems high to me. Note that a global expert elicitation estimated that 46% of salmonella in Africa was related to foodborne transmission.

Thank you for your comment. This figure of 86% (corrected from 89% within the text) is high, correct according to the Majowicz (2010). In order to put this figure into the African context we have added the following into the text to line 52:

‘This estimate is lower on the African continent, where forty-six percent of cases of illness caused by non-typhoidal *Salmonella* are attributed to exposure through the foodborne pathway (7).’

2. Line 94. What was the success rate of collecting samples from humans, as it looks like less than one per person per visit. Please present data on isolation rates by age, as we would expect the majority of salmonellae in young children.

Thank you for your comment. During the study we collected a stool sample from each human who had resided within the household perimeter for the last seven nights prior to the study team visit. No humans enrolled into the study were displaying clinical signs of *Salmonella* infection, therefore we were assessing for carriage isolates only. Humans were willing to donate stool for use within the project. In total 184 humans were enrolled to be part of the study, many of whom submitted samples during sample collection at more than one and up to three of the three visits to each of the households, rendering the total number of faecal samples collected from humans during the study to be 411. This was 100% of submissions which were requested, all were submitted within the required timeframe.

Below is data presenting isolation rate by age, as confirmed by PCR for the *ttr* gene.

Age	Total number sampled (n=411)	Total number PCR Salmonella positive (n=34)	Percentage Salmonella positive of samples collected
5 or less	43	1	2.3
6-17	115	8	7.0
18 and above	177	20	11.3
Unknown	76	5	6.6

While we would agree that we would expect the majority of *Salmonella* disease to occur in children, this study investigates carriage rates of *Salmonella* within the gut of healthy individuals. As you can see from the data in the table above, we have not found carriage rates of *Salmonella* to be highest within children. A recent study in Kenya has detected that carriage of *Salmonella* within healthy humans to be highest within the 15-24 years age bracket (8). The aim of this study was to sample stool from all ages present within each household to thoroughly investigate carriage and sharing of *Salmonella* within households.

3. Line 95. In terms of sampling from households, what were the reasons for such high variability in numbers of samples collected?

Thank you for your comment. As you would expect each household had different numbers of humans and animal species present compared to other households, and this frequently varied between individual households at each separate visit. We aimed to collect at least one sample from each human present and at least one from each species of animal also present in that household. Therefore, considering the composition of species within each household was often vastly different, there is a high variability in numbers of the samples collected from each individual household visit.

4. I can't understand the distribution of genomes between the two study sites. In lines 104-5, 87 samples from Ndirande and 146 from Chikwawa are positive. In lines 112-3, it states that there 32 genomes from Ndirande and 195 from Chikwawa.

How is there more genomes from Chikwawa than positive samples?

Thank you for your comment. This difference reflects the strategy taken to select *Salmonella* isolates to go forward for whole genome sequencing. We have added the following lines (122-125) to explain this:

'Whole genome sequencing was used to confirm the presence of *Salmonella* spp. We performed DNA extraction and whole genome sequencing of two-three *ttr* positive colony picks per sample to enable us to assess multi-serovar carriage and within-host diversity. We therefore sequenced 403 isolates in total, which were taken from a total of 214 samples.'

We have also clarified this in Suppl. Figure 2.

This method of selecting two-three isolates from each sample (eg Human A) for sequencing provides an explanation as to why more genomes from Chikwawa underwent whole genome sequencing than the number of positive *Salmonella* isolates detected within Chikwawa.

Were there more than one colony picks off positive plates in some samples?

Thank you for your comment. As mentioned in the comment above, two to three colony picks from positive plates were submitted for whole genome sequencing. This was to assess for within-host *Salmonella* diversity, which we did detect during this study.

This has been clarified in the methods on line 698;

'DNA of two-three frozen colonies per sample was quantified using a Qubit© (Thermo Fisher Scientific, MA, USA) and then sent for whole genome sequencing at the Wellcome Sanger Institute, UK.'

and updated in the outline of the Methods provided within Suppl. Figure 2.

If the different distribution relates to quality of sequencing, why was there such low quality in Ndirande?

Thank you for this comment.

This different distribution does not relate to the quality of the sequencing, rather the presence of more *Salmonella* within Chikwawa than in Ndirande. Our screening test using *ttr* suggested the presence of *Salmonella* in 233/2,080 samples, 87/965 (9.0%) of which were from Ndirande and 146/1,115 (13.1%) of which were from Chikwawa. As stated above, whole genome sequencing was carried out on two-three colony picks per sample to assess within host diversity and therefore in total DNA from 403 isolates was submitted for whole genome sequencing. However, subsequent whole genome sequencing only confirmed the presence of *Salmonella* in 227 isolates (194/227 (85.5%) were from Chikwawa and 33/227 (14.5%) were from Ndirande) which originated from 111 samples. In total 94/1,115 (8.4%) of samples collected from Chikwawa and 17/965 (1.8%) of samples collected in Ndirande contained *Salmonella*.

The vast majority of these isolates submitted for whole genome sequencing were confirmed as alternative bacteria, particularly *Proteus* spp., rather than poor quality *Salmonella* genomes. We found that the prevalence of *Salmonella* was lower in the urban area Ndirande than in the rural area, Chikwawa, nevertheless the samples remain representative of the study population.

This has been clarified in the text between lines 114-155:

‘In total, 2,080 individual samples were collected, 965 (46.4%) from Ndirande and 1,115 (53.6%) from Chikwawa. Samples were cultured and the presence of *Salmonella* was screened for by PCR for *ttr*. PCR for *ttr* was positive in 233/2,080 samples (11.2%) (Suppl. Figure 2, Suppl. Table 1-3). In total 87/965 (9.0%) of samples from Ndirande and 146/1,115 (13.1%) of samples from Chikwawa were positive for *Salmonella* spp..

Distribution of *Salmonella* genomes

Whole genome sequencing was used to confirm the presence of *Salmonella* spp. We performed DNA extraction and whole genome sequencing of two-three *ttr* positive colony picks per sample to enable us to assess multi-serovar carriage and within-host diversity. We therefore sequenced 403 isolates in total, which were taken from a total of 214 samples. After quality assessment of sequence data, 227 genomes were identified as *Salmonella* genomes, passing quality thresholds, and were subsequently included in our analysis (Figure 1A). These 227 genomes originated from 111 discrete samples.

On examination, this collection of 227 genomes contained a number of ‘identical’ bacterial isolates (i.e. identical bacterial isolates are detected within one sample, collected at the same household at the same timepoint). Identical isolates were de-duplicated from this collection, leaving a total of 131 individual isolates (an individual isolate is a single *Salmonella* isolate of each serovar or sequence type from each individual sample). These 131 individual isolates were collected from 111 samples, therefore more than one individual *Salmonella* isolate was identified from twenty of these samples (within-host diversity). The 111 samples originate from 81 animal stool samples (73.0%), 16 environment samples (14.4%) and 14 human stool samples (12.6%). At least one *Salmonella* genome was generated from 25/30 households in the study; 14/15 (93.3%) households sampled in Chikwawa and 11/15 (73.3%) households sampled in Ndirande (Figure 1A).

In total 94/1,115 (8.4%) of samples collected from Chikwawa and 17/965 (1.8%) of samples collected in Ndirande contained *Salmonella*.

5. Line 132, this multi-panel figure is a bit confusing as readers may think that the legend for part a relates to the remaining panels. I suggest separating part a and b. I had a hard time distinguishing the 64 different sequence types in the part b of the figure.

Thank you for your comment. We have separated part a) and b) of the legend in order to help improve the clarity. We do understand that it is hard to distinguish 64 different sequence types from Part b of the figure. The main aim of the Figure is to provide an impression of the huge diversity of *Salmonella* encountered within each household and sample type. We do feel that this is has been achieved.

6. In line 151, the authors describe using Chi Square tests for testing differences between categories. For all statistical tests, it is important to consider using Fischer Exact if some of the expected values in cells are less than five.

Thank you for your comment. Where appropriate, we have used Fisher's exact test, i.e. when the values were less than five. In the instances mentioned within the text of the paper, Chi squared tests were statistically appropriate.

7. Figure 3. Suggest using something other than a tree to denote environment, although I know this is tricky.

Thank you very much for this suggestion. We have not made this change, but instead we have clarified the legend to explain this by adding the following text to line 453:

'Human silhouette indicates genomes initially collected from humans. Animal silhouettes represent *salmonellae* initially collected from animals. The tree silhouette represents salmonellae collected from within the environment of each household perimeter. Each silhouette represents the sample environment or animal species as a whole.'

8. Line 416. I don't think the term 'the Global North' is helpful to distinguish groups of countries. Why not talk about 'industrialized' settings or 'high-income' countries, as you are really trying to say that proximity to animals leads to shared salmonellae. This definitely happens in high-income countries where close proximity occurs.

Thank you for your comment. We have changed line 503 to read 'more industrialised agricultural settings' settings rather than Global-North.

9. Line 421. I really don't like claims of primacy for things like this, as it doesn't really make it more important and it may not be true. There has been a lot of work done on sharing strains of salmonellae in the urban environment. In Australia, investigators found a high prevalence of shared salmonellae in the Darwin urban environment, which is tropical in nature. See:

a. Williams S, Markey P, Harlock M, Binns P, Gaggin J, Patel M. Individual and household-level risk factors for sporadic salmonellosis in children. J Infect. 2016 Jan;72(1):36-44. doi: 10.1016/j.jinf.2015.09.014. Epub 2015 Sep 28. PMID: 26416475.

b. Williams S, Patel M, Markey P, Muller R, Benedict S, Ross I, Heuzenroeder M, Davos D, Cameron S, Krause V. Salmonella in the tropical household environment--Everyday, everywhere. J Infect. 2015 Dec;71(6):642-8. doi: 10.1016/j.jinf.2015.09.011. Epub 2015 Sep 28. PMID: 26416474.

Thank you very much for this comment and thank you for these interesting citations. We have altered the associated sentence in the following manner on line 508, and so removed the claim of primacy:

‘This study investigates sharing of *Salmonella* within households located in both rural and urban environments in sub-Saharan Africa.’

10. You don’t really talk much about the most likely animal reservoirs of *S. salamae*. Do you think there were unsampled animal reservoirs, such as snakes, frogs and other lizards in the study? Is this worth further investigation.

Thank you for this comment. It would certainly be worth further investigation of these other potential reservoirs, but it is not possible for us to comment as they did not form part of our sample frame.

11. Line 458. Do you know if *S. salamae* causes much human disease in terms of gastroenteritis and invasive infections in Malawi?

Currently there are no data on this and so we have not speculated on this in the paper.

12. Line 464. Throughout the manuscript the authors talk about ‘transmission’. Here it mentions ‘transmission between households’. In reality, this study is not documenting transmission, but point prevalence. I feel that the use of transmission is too strong, as it implies that one gave it to the other, which isn’t really a correct interpretation. The same strain was isolated from multiple households or species.

Thank you for this comment, we agree that this is an important point. We have changed the language to discuss putative sharing and flux of *Salmonella*, rather than direct transmission. The similarity of the genomes does raise the possibility for transmission, but we agree that we cannot document absolutely that this has occurred.

13. Line 475. I think that the sharing of strains between dogs and poultry isn’t that interesting, it is expected.

We respectfully disagree, mainly as sharing between these species would imply movement from livestock to a domestic pet species, and more so to a species (the dog) that is known to roam widely between households and the broader environment and therefore potentially place a role in distributing a strain. More so, expected or not, it is important to present evidence to support that notion, which we have.

14. Line 483. The authors mention that animal faeces removal and hand hygiene is ‘profoundly important’ in reducing spread of salmonellae. I think it is important not to overplay the fact that shared strains in different hosts were identified, but the connection to disease and transmission is less clear. This also begs the question about cleanliness of food preparation areas and food storage, which are important.

Thank you for this comment. In order to clarify that the link between transmission and disease is not clear we have added in the sentence ‘The connection and relationship between carriage and disease of non-typhoidal *Salmonella* is not yet fully clear across all serovars in all species.’ to line 521.

Within the paragraph starting on line 602 we are specifically referring to practises to improve biosecurity in relation to animal husbandry practises. We would certainly agree that many different aspects of environmental health, including cleanliness of food preparation areas and food storage, are important to consider in efforts to reduce the spread of bacteria. We have added the following sentence to line 608 to clarify this:

‘A range of environmental health practises have been shown to be important to reduce the transmission of bacteria and antimicrobial resistance determinants ^{32,54,55.}’

15. Supplementary information: I note a high prevalence of Salmonella Gaminara. Is this serotype known to cause disease in humans?

Outbreaks of disease due to *S. Gaminara* in humans have sporadically been reported, particularly in the USA (9,10).

Are any of the other serotypes less pathogenic to humans? It would be good to display this information by host species. I also note a high diversity of *S. salamae* serotypes as well.

Thank you for this comment. It is not clear, and unfortunately beyond the scope of this study to investigate further.

References

1. Musicha P, Cornick JE, Bar-Zeev N, French N, Masesa C, Denis B, et al. Trends in antimicrobial resistance in bloodstream infection isolates at a large urban hospital in Malawi (1998–2016): a surveillance study. *Lancet Infect Dis*. 2017 Oct 1;17(10):1042–52.
2. Koolman L, Prakash R, Diness Y, Msefula C, Nyirenda TS, Olgemoeller F, et al. Case-control investigation of invasive Salmonella disease in Malawi reveals no evidence of environmental or animal transmission of invasive strains, and supports human to human transmission. *PLoS Negl Trop Dis*. 2022 Dec 12;16(12):e0010982.
3. Post AS, Diallo SN, Guiraud I, Lompo P, Tahita MC, Maltha J, et al. Supporting evidence for a human reservoir of invasive non-Typhoidal Salmonella from household samples in Burkina Faso. *PLoS Negl Trop Dis*. 2019 Oct 14;13(10):e0007782.
4. Koolman L, Prakash R, Diness Y, Msefula C, Nyirenda TS, Olgemoeller F, et al. Case-control investigation of invasive Salmonella disease in Africa – comparison of human, animal and household environmental isolates find no evidence of environmental or animal reservoirs of invasive strains [Internet]. *medRxiv*; 2022 [cited 2022 May 7]. p. 2022.01.31.22270114. Available from: <https://www.medrxiv.org/content/10.1101/2022.01.31.22270114v1>
5. Falay D, Hardy L, Tanzito J, Lunguya O, Bonebe E, Peeters M, et al. Urban rats as carriers of invasive Salmonella Typhimurium sequence type 313, Kisangani, Democratic Republic of Congo. *PLoS Negl Trop Dis*. 2022 Sep 6;16(9):e0010740.
6. Majowicz SE, Musto J, Scallan E, Angulo FJ, Kirk M, O’Brien SJ, et al. The global burden of nontyphoidal Salmonella gastroenteritis. *Clin Infect Dis Off Publ Infect Dis Soc Am*. 2010 Mar 15;50(6):882–9.

7. Hald T, Aspinall W, Devleeschauwer B, Cooke R, Corrigan T, Havelaar AH, et al. World Health Organization Estimates of the Relative Contributions of Food to the Burden of Disease Due to Selected Foodborne Hazards: A Structured Expert Elicitation. *PLOS ONE*. 2016 Jan 19;11(1):e0145839.
8. Muthumbi EM, Mwanzu A, Mbae C, Bigogo G, Karani A, Mwarumba S, et al. The epidemiology of fecal carriage of nontyphoidal *Salmonella* among healthy children and adults in three sites in Kenya. *PLoS Negl Trop Dis*. 2023 Oct 26;17(10):e0011716.
9. Cornwell CMM. FATE OF *SALMONELLA MONTEVIDEO*, *SALMONELLA GAMINARA*, AND *SALMONELLA POONA* IN HOMEMADE UNPASTEURIZED FRUIT AND VEGETABLE JUICES.
10. Lang BY, Varman M, Reindel R, Hasley BP. *Salmonella Gaminara* Osteomyelitis and Septic Arthritis in an Infant With Exposure to Bearded Dragon. *Infect Dis Clin Pract*. 2007 Sep;15(5):348–50.